# NF-MKV Net: A Constraint-Preserving Neural Network Approach to Solving Mean-Field Games Equilibrium

## Abstract

Neural network-based methods for solving Mean-Field Games (MFGs) equilibria have garnered significant attention for their effectiveness in high-dimensional problems. However, many algorithms struggle with ensuring that the evolution of the density distribution adheres to the required mathematical constraints. This paper investigates a neural network approach to solving MFGs equilibria through a stochastic process perspective. It integrates process-regularized Normalizing Flow (NF) frameworks with state-policy-connected time-series neural networks to address McKean-Vlasov-type Forward-Backward Stochastic Differential Equation (MKV FBSDE) fixed-point problems, equivalent to MFGs equilibria. First, we reformulate MFGs equilibria as MKV FBSDEs, embedding the density distribution into the equation coefficients within a probabilistic framework. Neural networks are then designed to approximate value functions and gradients derived from these equations. Second, we employ NF architectures, a class of generative neural network models, and impose loss constraints on each density transfer function to ensure volumetric invariance and time continuity. Additionally, this paper presents theoretical proofs of the algorithm's validity and demonstrates its applicability across diverse scenarios, highlighting its superior effectiveness compared to existing methods.

## 1 Introduction

Mean-Field Games (MFGs), introduced independently by Lasry & Lions (2007) and Huang et al. (2006), provide a robust framework for addressing large-scale multi-agent problems. MFGs are widely applied in domains such as autonomous driving, social networks, crowd management, and power systems.

Neural network-based algorithms have recently been employed to solve MFGs equations due to their ability to handle high-dimensional problems effectively. For example, Lin et al. (2021) reformulated MFGs as a generative adversarial network (GAN) training problem, and Ruthotto et al. (2020) introduced a Lagrangian-based approach to approximate agent states through sampling. Additionally, Chen et al. (2023) applied reinforcement learning and neural networks to model distributions and address value functions. Most approaches focus on optimizing the loss term in MFGs coupled equations using sampled agents, but they often neglect density dynamics, leading to challenges in representing continuous state density distributions.

Carmona et al. (2018) introduced a stochastic process perspective on MFGs, leveraging McKean-Vlasov Forward-Backward Stochastic Differential Equations (MKV FBSDEs) to address MFGs equilibria, and explored numerical methods for solving them. Achdou & Lauriere (2015) proposed simplified MFGs models for pedestrian dynamics and demonstrated them with numerical simulations. Ren et al. (2024) studied multi-group MFGs by solving asymmetric Riccati differential equations and established sufficient conditions for the existence and uniqueness of optimal solutions. However, existing MKV FBSDE methods are often limited to linear-quadratic MFGs, where distributions are simplified to the expectation of agents' states, rather than full distribution functions, in nonlinear settings. Recently, Huang et al. (2023) proposed a data-driven Normalizing Flow (NF) approach to solve distribution-involved optimal transport stochastic problems. Nevertheless, con-

straints from MFGs process dynamics, terminal loss, and equation coupling limit the availability of NF frameworks for high-dimensional MKV FBSDEs.

In summary, solving the MFGs equilibrium reduces to addressing equivalent stochastic fixed-point problems that incorporate distribution flows. NF-MKV Net is proposed to solve the MKV FBSDEs problem by coupling the process-regularized NF and state-policy-connected time series neural networks. The enhanced NF framework models flows of probability measures, constructing a density distribution flow that satisfies volumetric invariance at each time step. State-policy-connected time series neural networks, grounded in MKV FBSDEs, model relationships between time-step value functions and approximate their gradients, enabling solutions in a time-consistent manner. Using the coupled frameworks, the fixed-point distribution flow equivalent to the MFGs equilibrium can be determined while ensuring mathematical constraints are satisfied.

**Contributions**: The main contributions and results are summarized as follows:

- NF-MKV Net is proposed to solve MKV FBSDEs, which are equivalent to MFGs equilibrium, from a stochastic process perspective. By integrating process-regularized NFs and state-policy-connected time series neural networks into a coupled framework with alternating training, the method adheres to volumetric invariance and time-continuity constraints.

- Process-regularized NF frameworks are designed to model probability measure flows by enforcing loss constraints on each density transfer function, ensuring volumetric invariance at each time step.

- State-policy-connected time series neural networks, built upon MKV FBSDEs, capture the relationships between time-step value functions and approximate their gradients, enabling time-consistent solutions.

- The method demonstrates applicability in traffic flow, low- and high-dimensional crowd motion, and obstacle avoidance problems. Additionally, it satisfies mathematical constraints better than existing neural network-based approaches.

## 2 CONNECTIONS AMONG MFG, MKV, AND NF

### 2.1 MFGS ↔ MCKEAN-VLASOV FBSDE

We now formalize the MFGs problem without considering common noise. For this purpose, we start with a complete filtered probability space $(\Omega, \mathcal{F}, \mathbb{F} = (\mathcal{F}_t)_{0 \leq t \leq T}, \mathbb{P}))$ the filtration $\mathbb{F}$ supporting a $d-$dimensional Wiener process $\mathbf{W} = (W_t)_{0 \leq t \leq T}$ with respect to $\mathbb{F}$ and an initial condition $\xi \in L^2(\Omega, \mathcal{F}_0, \mathbb{P}; \mathbb{R}^d)$. This MFGs problem can be described as:

(i) For each fixed deterministic flow $\boldsymbol{\mu} = (\mu_t)_{0 \leq t \leq T}$ of probability measures on $\mathbb{R}^d$, solve the standard stochastic control problem:

$$\inf_{\alpha \in \mathbb{A}} J^\mu(\alpha) \quad \text{with} \quad J^\mu(\alpha) = \mathbb{E}[\int_0^T f(t, X_t^\alpha, \mu_t, \alpha_t)dt + g(X_T^\alpha, \mu_T)], \tag{1}$$

subject to

$$\begin{cases} dX_t^\alpha = b(t, X_T^\alpha, \mu_t, \alpha_t)dt + \sigma(t, X_T^\alpha, \mu_t, \alpha_t)dW_t, \quad t \in [0, T], \\ X_0^\alpha = \xi \end{cases} \tag{2}$$

(ii) Find a flow $\boldsymbol{\mu} = (\mu_t)_{0 \leq t \leq T}$ such that $\mathcal{L}(\hat{X}_T^{\boldsymbol{\mu}}) = \mu_t$ for all $t \in [0, T]$, if $\hat{X}^{\boldsymbol{\mu}}$ is a solution of the above optimal control problem.

We can see that the first step provides the best response of a given player interacting with the statistical distribution of the states of the other players if this statistical distribution is assumed to be given by $\mu_t$. In contrast, the second step solves a specific fixed point problem in the spirit of the search for fixed points of the best response function.

Usually, the solution of MFGs is transformed into a set of coupled partial differential equations, namely the HJB-FPK equations, which respectively describe the evolution of the value function of the representative element and the density evolution of the group, as shown below:

$$
\begin{aligned}
& -\partial_t u - \nu \Delta u + H(x, \nabla u) = f(x, \mu) && \text{(HJB)} \\
& \partial_t \mu - \nu \Delta \mu - \text{div}(\mu \nabla_p H(x, \nabla u)) = 0 && \text{(FPK)} \\
& \mu(x, 0) = \mu_0, \quad u(x, T) = g(x, \mu(\cdot, T))
\end{aligned}
\tag{3}
$$

in which, $u : \mathbb{R}^n \times [0, T] \to \mathbb{R}$ is the value function to guide the agents make decisions; $H : \mathbb{R}^n \times \mathbb{R}^n \to \mathbb{R}$ is the Hamiltonian, which describes the physics energy of the system; $\mu(\cdot, t) \in \mathcal{L}(\mathbb{R}^n)$ is the distribution of agents at time t, $f : \mathbb{R}^n \times \mathcal{L}(\mathbb{R}^n) \to \mathbb{R}^n$ denotes the loss during process; and $g : \mathbb{R}^n \times \mathcal{L}(\mathbb{R}^n) \to \mathbb{R}^n$ is the terminal condition, guiding the agents to the final distribution.

Let assumption **MFGs Solvability HJB** (as shown in Appendix A.1) be in force. Then, for any initial condition $\xi \in L^2(\Omega, \mathcal{F}_0, \mathbb{P}; \mathbb{R}^d)$, the *McKean-Vlasov FBSDEs*:

$$
\begin{cases}
dX_t = b(t, X_t, \mathcal{L}(X_t), \hat{\alpha}(t, X_t, \mathcal{L}(X_t), \sigma(t, X_t, \mathcal{L}(X_t))^{-1\dagger} Z_t)) dt + \sigma(t, X_t, \mathcal{L}(X_t)) dW_t \\
dY_t = -f(t, X_t, \mathcal{L}(X_t), \hat{\alpha}(t, X_t, \mathcal{L}(X_t), \sigma(t, X_t, \mathcal{L}(X_t))^{-1\dagger} Z_t)) dt + Z_t \cdot dW_t
\end{cases}
\tag{4}
$$

for $t \in [0, T]$, with $Y_T = g(X_T, \mathcal{L}(X_T))$ as terminal condition, is solvable. Moreover, the flow $(\mathcal{L}(X_T))_{0 \le t \le T}$ given by the marginal distributions of the forward component of any solution is an equilibrium of the MFGs problem associated with the stochastic control problem Eq.(1).

We consider the optimal control step of the formulation of the MFGs problems described earlier. Probabilists have a two-pronged approach to these optimal control problems. We consider that the input $\boldsymbol{\mu} = (\mu_t)_{0 \le t \le T}$ is deterministic and fixed to search the optimal reaction decision. Then, after the result of fixed decision, the optimal flows of probability measure can be solved. Alternately seeking for the optimal control, the MFGs equilibrium can be finally derived.

## 2.2 MFGs ↔ NF

NNormalizing Flows (NF), introduced by Tabak & Vanden-Eijnden (2010), enable exact computation of data likelihood through a sequence of invertible mappings. A key feature of NF is its use of arbitrary bijective functions, achieved through stacked reversible transformations. The flow model $R$ consists of a sequence of reversible flows, expressed as $R(x) = r_1 \circ r_2 \circ \cdots \circ r_L(x)$, where each $r_i$ has a tractable inverse and Jacobian determinant.

Our algorithm leverages the volume-preserving property of NF, aligning with the consistency of density flow in MFGs during evolution. This principle is essential for constructing the density flow in the MFGs model.

The connection between MFGs and NF provides inherent advantages. For example, in MFGs, the initial distribution is often represented in a simple analytical form. This parallels NF's approach, where a simple initial distribution transforms into a more complex one for density estimation. Additionally, one advantage of NF over other generative models is its preservation of total density during transformation, consistent with the MFGs requirement $\int \mu dx = 1$. A challenge, however, is that MFGs, unlike Optimal Transport (OT), lacks both initial and terminal density distributions. In MFGs, only an initial distribution exists, and the terminal condition is governed by the terminal value function $g(x, \mu(\cdot, T))$. This complicates framing the problem as a complete density evolution problem.

To address this, we idealize the MFGs model with the assumption that the terminal value function corresponds to an explicitly solvable optimal terminal density. For example, in trajectory planning problems, the terminal value function is often related to the destination, such as $g(x, \mu) = \int_{\mathbb{R}^d} -e^{-||x - x_T||^2} d\mu(x)$. In such cases, we assume the optimal terminal distribution is $\mu(x_T) = 1$, enabling the MFGs problem to be framed with initial and terminal densities. We will show that this assumption is reasonable.

# 3 METHODOLOGY: NF-MKV NET

We propose NF-MKV Net, an alternately trained model combining NF and McKean-Vlasov Forward-Backward Stochastic Differential Equations (MKV FBSDEs), to address MFGs equilibrium problems. The main advantage of MKV FBSDEs is their ability to capture both optimization and interaction components in a single coupled FBSDE, eliminating the need for separate references to Hamilton-Jacobi-Bellman (HJB) and Fokker-Planck-Kolmogorov (FPK) equations. NF generates flows represented by neural networks, constraining each density transfer function to define density distributions at specific times. The density flow from NF couples with the value function of the MFG. The value function constrains the neural network generating the flow via the HJB equation, while its gradient update depends on the current marginal distribution flow.

First, we reformulate the stochastic equations of MFGs using MKV FBSDEs and approximate value function gradients with neural networks, effectively addressing the curse of dimensionality in traditional numerical methods. Second, to address the distribution-coupled challenge, we use NF architectures to model agents' state density distributions, alternately training the unknown transition processes with value functions. Figure (1) illustrates the framework of NF-MKV Net.

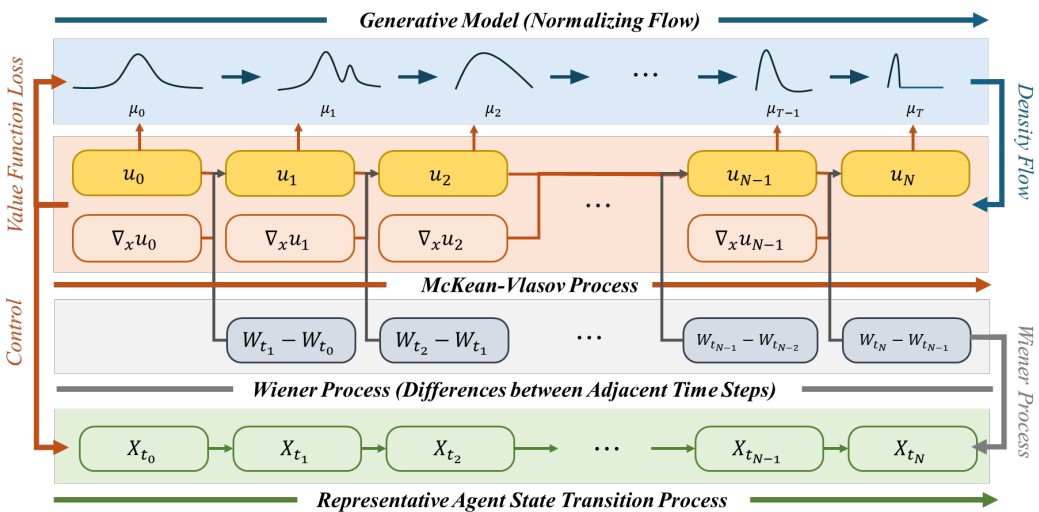

Figure 1: Framework Diagram of the NF-MKV Net

## 3.1 MODELING VALUE FUNCTION WITH MKV FBSDES

We consider a general class of MFGs problem associated with the stochastic control problem (1), the relative FBSDEs can be represented as in (4), with initial condition $\xi \in L^2(\Omega, \mathcal{F}_0, \mathbb{P}; \mathbb{R}^d)$ and terminal condition $Y_T = g(X_T, \mathcal{L}(X_T))$.

Then the solution of Eq.(3) satisfies the following FBSDE:

$$u(x,t) - u(x,0) = -\int_0^t f(s, X_s, \mathcal{L}(X_s), \hat{\alpha}(s, X_s, \mathcal{L}(X_s), \sigma(t, X_s, \mathcal{L}(X_s))^{-1\dagger} Z_s)) ds + \int_0^t Z_s dW_s. \tag{5}$$

We apply a temporal discretization to Eq.(4). Given a partition of the time interval $[0, T] : 0 = t_0 < t_1 < \cdots < t_N = T$, we consider the simple Euler scheme for $n = 1, \cdots, N - 1$

$$X_{t_{n+1}} - X_{t_n} \approx b(t, X_T^\alpha, \mu_t, \alpha_t)\Delta t_n + \sigma(t, X_T^\alpha, \mu_t, \alpha_t)\Delta W_n, \tag{6}$$

and

$$u(x, t_{n+1}) - u(x, t_n) \approx -f(s, X_s, \mathcal{L}(X_s), \hat{\alpha}_t(\cdot)^{-1\dagger} Z_s))\Delta t_n + [\partial_x u(x,t)]^T \sigma \Delta W_n, \tag{7}$$

where

$$\Delta t_n = t_{n+1} - t_n, \quad \Delta W_n = W_{t_{n+1}} - W_{t_n}. \tag{8}$$

The key in modeling the above FBSDEs is to approximate the value function $x \mapsto u(0,x)$ at $t = t_0$, which is $u(0,x) \approx u(0,x|\theta_0)$; while approximating the function $x \mapsto [\partial_x u(x,t)]^T \sigma$ at each time step $t = t_n$ through a multi-layer feedforward neural network

$$x \mapsto [\partial_x u(x,t)]^T \sigma \approx [\partial_x u(x,t)|\theta_n]^T \sigma, \tag{9}$$

which represents the adjoint variable $Z_t$ of the value function of the element, which is expressed as the product of the gradient and the random variable in the MFGs optimization problem.

Subsequently, all value functions are connected by summing Eq.(7) over $t$. The network uses the generated density flows $\boldsymbol{\mu}$ and $W_{t_n}$ as inputs and produces the final output $\hat{u}$, approximating $u(x,T) = g(x, \mu(\cdot, T))$. This approximation defines the expected loss function by comparing the maximum likelihoods of the two functions to minimize the difference for $\{x_i\}_{i=1}^N \sim z = \hat{\mu}_T$:

$$l_{MKV} = -\log \mathbf{p}(g(z, \hat{\mu}_T)|\hat{u}(\theta, z)) = -\frac{1}{N} \sum_{i=1}^N \log \mathbf{p}(g(x_i, \hat{\mu}_T)|\hat{u}(\theta, x_i)). \tag{10}$$

In summary, we reformulate MFGs as MKV FBSDEs (Eq.(5)) and discretize time to establish the relationship between value functions at each time $t$ (Eq.(7)), connecting them via the adjoint variable $Z_t$. Next, we parameterize $Z_t$ and $u_0$, using these relationships to link $u_T$ with the terminal condition $g(x, \mu(\cdot, T))$ for maximum likelihood estimation (Eq.(10)) to minimize the final MFGs loss.

## 3.2 Modeling density distribution with NF

Typically, NF methods prioritize density estimation results. In contrast, our approach emphasizes the NF density evolution process, constraining each layer to align with the density evolution in MFGs.

In an NF model, If $r_i$ are differentiable and reversible functions, we usually express it like:

$$\begin{array}{llll} \text{(Normalizing)} & \mathbf{r} = r_1 \circ r_2 \circ \cdots \circ r_N, & \mathbf{p}_{\mu_0}(X) = & \mathbf{p}_{\mu_T}(\mathbf{r}(X))|\det D\mathbf{r}(X)| \\ \text{(Construct)} & \mathbf{s} = s_N \circ s_{N-1} \circ \cdots \circ s_1, & \mathbf{p}_{\mu_T}(X) = & \mathbf{p}_{\mu_0}(\mathbf{s}(X))|\det D\mathbf{r}(X)|^{-1} \end{array} \tag{11}$$

To simplify the explanation, we consider a one-dimensional model. During training, $r_i(x)$ is represented as a neural network, $r_i(x; \phi)$. Multiple $r_i$ are combined to obtain the desired function $f$. Typically, training minimizes the negative log-likelihood loss between the final estimated density and the dataset, expressed as

$$L(x) = -\log \mathbf{p}_{\mu_T}(x) = -\log \mathbf{p}_{\mu_0}(\mathbf{r}^{-1}(x)) - \log |\det D\mathbf{r}^{-1}(x)| \tag{12}$$

in which there is no need to consider the loss of each $r_i$ in the process.

Discretizing the NF construction process reveals that each function corresponds to Euler time discretization. Thus, each sub-function $r_i$ transforms the group density $\mu_i$ in MFGs into $\mu_{i+1}$. This series of reversible flows represents the time evolution process, which results in that

$$\mu_0 \xrightarrow{\mathbf{r}(x)} \mu_T \quad \Leftrightarrow \quad \mu_0 \xrightarrow{r_{t_1}(x)} \mu_{t_1} \xrightarrow{r_{t_1}(x)} \mu_{t_2} \xrightarrow{r_{t_2}(x)} \cdots \xrightarrow{r_{t_{N-1}}(x)} \mu_{t_N} \tag{13}$$

Additionally, as each sub-function is implemented as a neural network, the loss at each time step can be used to constrain and optimize the sub-functions. The density $\mu_t$ can be expressed as $\mu_{t_n} = \mathbf{r}_{1,2,\cdots,n} \circ \mu_0$, where $\mathbf{r}_{1,2,\cdots,n} = r_{t_n} \circ r_{t_{n-1}} \circ \cdots \circ r_{t_1}$.

Our way of modeling the limited is to approximate the crowd density of each layer

$$\mu_0 \mapsto \mathbf{r}_{1,2,\cdots,n}(x) \circ \mu_0(x) \approx \mathbf{r}_{1,2,\cdots,n}(x; \boldsymbol{\Phi}) \circ \mu_0 \tag{14}$$

at each time step $t = t_n$ through a NF model consisting of multiple layers of Masked Autoregressive Flow (MAF) and Permute parameterized by $\phi$.

To train the NF, the first step is computing the process loss $l_{\text{HJB}}$. At each time step $t = t_n$, the MFGs system satisfies the HJB equation. Section 3.1 describes the value function and its gradient. Thus, samples $x_{i=1}^M$ from $\mu_t$ at each time step can be used in the HJB equation to compute the loss,

$$l_{\text{HJB}} = \frac{1}{N} \frac{1}{M} \sum_{n=1}^N \sum_{i=1}^M \|\partial_t u(x_i, t_n) + \nu \Delta u(x_i, t_n) - H(\nabla_x u(x_i, t_n)) + f(x_i, t_n)\|^2 \tag{15}$$

where

$$\left(\{x_i\}_{i=1}^M, t_n\right) \sim \mu_{t_n} \approx \mu_{t_n}(\phi) = r_{t_n}(x; \phi_n) \circ r_{t_{n-1}}(x; \phi_{n-1}) \circ \cdots \circ r_{t_1}(x; \phi_1) \circ \mu_0. \quad (16)$$

Additionally, the NF method must match the terminal density condition, so the terminal loss $l_T$ is also included in the loss calculation. If the terminal condition $g$ is explicitly defined, the corresponding optimal density $\hat{\mu}_T(\phi) = \mathbf{f}(x; \Phi) \circ \mu_0$ can serve as the target distribution for NF. The negative log-likelihood between $\hat{\mu}T(\phi)$ and $\mu_T$ generated by NF is used to compute the terminal loss $l$T. For $x_{i_{i=1}}^N \sim \hat{\mu}_T(\Phi)$:

$$l_T = -\log \mathbf{p}(\mu_T | \hat{\mu}_T(\Phi)) = -\frac{1}{N} \sum_{i=1}^N \log \mathbf{p}(\mu_T | \hat{\mu}_T(\Phi, x_i)). \quad (17)$$

In summary, NF, as a generative model, can construct intermediate function compositions and distributions without direct data use, relying solely on the initial distribution $\mu_0$ and the terminal distribution $\mu_T$, while preserving density consistency. We first compute the terminal density distribution $\mu_T$ explicitly and construct an NF to transition from $\mu_0$ to $\mu_T$. The losses $l_{\text{HJB}}$ (Eq. 15) and $l_T$ (Eq. 17) constrain NF evolution, ensuring the flow density aligns with the control objectives.

### 3.3 COUPLING TWO PROCESSES

Two processes can be coupled and trained alternately. As NF is a generative model, it can first generate a set of flow density evolution functions along with the corresponding density distributions at each time step. This generated set of density distributions is fixed as the marginal distribution to optimize the value function and gradient under the MKV FBSDE framework. Once the optimal value function for this marginal distribution flow is obtained, it is fixed to update each $r_n$ and its corresponding $\mu_{t_n}$ in the NF evolution process. This continues until the optimal density distribution flow under the current value function is achieved. This iterative coupled training continues until convergence. Algorithm (1) presents the pseudo-code of the model.

---

**Algorithm 1** NF-MKV Net

**Require:** $\sigma$ diffusion parameter, $g$ terminal cost, $H$ Hamiltonian, $f$ process loss, $\mu_0$ initial density
**Ensure:** $\boldsymbol{\mu} = (\mu_t)_{0 \le t \le T}$ density flow, $\mathbf{u} = (u_t)_{0 \le t \le T}$, value function
    $\mu_T \leftarrow \arg\max_\mu g(x, \mu(x, T))$
    Generating NF $\{\mu_{t_n}(\phi_n)\}_{n=1}^N$ from $\mu_0$ to $\mu_T$
    **while** not converged **do**
        **Train** $u(0, x|\theta_0)$ and $[\partial_x u(x, t)|\theta_n]^T \sigma$ for $n = 1, 2, \cdots, N$:
        Sample batch $\left(\{x_i\}_{i=1}^M, t_n\right) \sim \mu_{t_n}$ for $n = 1, 2, \cdots, N$
        Sample Winner Process $\{W_{t_n}\}_{n=1}^N \sim \mathcal{N}(0, \sigma^2)$
        $l_{MKV} \leftarrow -\frac{1}{N} |g((x_i, T), \mu_T)) - \hat{u}(\{(x_{i,n}, t_n)\}), \{W_{t_n}\}_{n=1}^N|^2$
        Back-propagate the loss $l_{MKV}$ to $\theta$ weights.
        **Train** $r_n(\phi_n)$ for $n = 1, 2, \cdots, N$:
        Sample batch $\left(\{x_i\}_{i=1}^M, t_n\right) \sim \mu_{t_n}(\phi_n)$ for $n = 1, 2, \cdots, N$
        $l_{\text{HJB}} \leftarrow \frac{1}{N} \frac{1}{M} \sum_{n=1}^N \sum_{i=1}^M \|\partial_t u(x_i, t_n) + \nu \Delta u(x_i, t_n) - H(\nabla_x u(x_i, t_n)) + f(x_i, t_n)\|^2$
        Sample batch $\{x_i\}_{i=1}^N \sim \hat{\mu}_T(\Phi)$
        $l_{\text{T}} \leftarrow -\frac{1}{N} \sum_{i=1}^N \log \mathbf{p}(\mu_T | \hat{\mu}_T(\Phi, x_i))$
        Back-propagate the loss $l_{\text{NF}} = l_{\text{HJB}} + l_T$ to $\phi$ weights.
    **end while**

---

## 4 NUMERICAL EXPERIMENT

We apply NF-MKV Net to MFGs instances and present the numerical results in two parts. The first part demonstrates NF-MKV Net as an effective method for solving MFGs equilibrium involving density distributions. The second part highlights the accuracy of NF-MKV Net in comparison to other algorithms.

### 4.1 SOLVING MFGS WITH NF-MKV NET

This section presents three examples of solving MFGs using NF-MKV Net, demonstrating its applicability to traffic flow problems, low- and high-dimensional crowd motion problems, and scenarios with obstacles.

#### 4.1.1 EXAMPLE 1: MFGS TRAFFIC FLOW CONTROL

A series of numerical experiments in MFGs Traffic Flow Control explore the dynamics of MFGs, focusing on autonomous vehicles (AVs) navigating a circular road network. The traffic flow scenario is formulated as an MFGs problem involving density distribution and the value function.

The initial density is defined on the ring road, where the state $x$ represents the AVs' position. The state transfer function is $dx = vdt + \sigma dW_t$, and the process constraint is $f(x) = \frac{1}{2}(1 - \mu - b)^2$. The Hamiltonian is defined as $H(x, p, t) = f(p, \mu) + pu_x$, leading to the optimal control $u^* = \arg\min_p(f(p, \mu) + pu_x)$. In the finite time domain problem, the terminal value function $u_T$ of the AVs system is constrained at $t = T$. It is assumed that AVs have no preference for their locations at time $T$, i.e., $u(x, T) = 0$. In the MFGs traffic flow problem, the terminal value function $u_T$ can be solved explicitly as $\mu_T(x) = 1, \forall x \in (0, 1)$, satisfying the model's assumptions.

Without loss of generality, we define the time interval as $[0, 1]$ and set the initial density $\mu_0$ at $t = 0$. To verify the volumetric invariance of the density distribution discussed in our study, we selected initial density functions satisfying $\int_0^1 \mu_0(x)dx$. Four different initial densities were selected, each with a distinct diffusion coefficient $\sigma$ for the Wiener process. NF-MKV Net was then employed to solve for equilibrium, verifying the proposed algorithm's applicability.

Results in Fig.(2) indicate that the agent distribution, regardless of initial density $\mu_0$ or drift term $\sigma$, converges to the equilibrium $\mu(x, T) = 1$. Comparing NF-MKV Net with the numerical solution shows errors below $10^{-3}$, demonstrating the algorithm's effectiveness in solving traffic flow problems. The results illustrate the evolution of the density distribution $\mu(x)$ over time $t$ and the log errors compared to the noise-free numerical method.

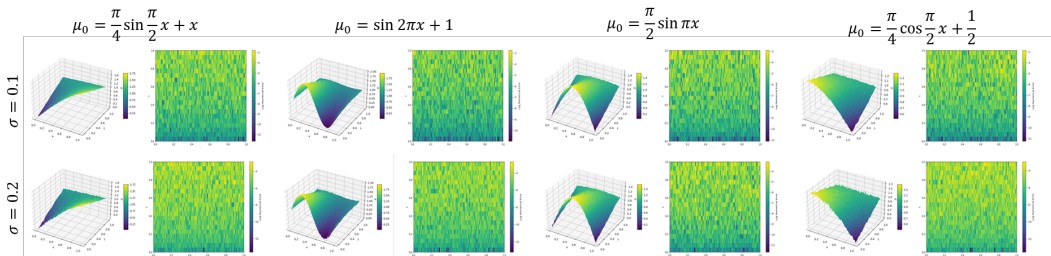

Figure 2: NF-MKV Net solutions and numerical error of MFGs traffic flow with various initial density distribution and diffusion coefficients

#### 4.1.2 EXAMPLE 2: MFGS CROWD MOTION

In this example, a dynamically formulated MFGs problem, the Crowd Motion problem, is constructed in dimensions $d = 2$ and $d = 50$ to demonstrate the applicability of NF-MKV Net. We set the problems as in Eq.(3) with parameter:

$$f(x, \mu) = \int_{\mathbb{R}^n} \mathbf{e}^{-|x-\hat{x}|^2} d\mu(\hat{x}), \qquad H(x, p, t) = |p|^2 + f(x, \mu, t),$$

$$\mu(x, 0) = \mu_0(x), \qquad\qquad u(x, T) = \int_{\mathbb{R}^n} |x - x_T|^2 d\mu(x) \tag{18}$$

**d = 2 Crowd Motion.** Here, $\sigma = \sqrt{2}$ is used with 20 time steps in the dynamics process, and the initial distribution is set as $\mu_0(x) = \mathcal{N}((-2, -2), (1^2, 1^2))$. To reach the goal point, we set $x_T = (2, -2)$ which means the terminal condition is $g(x, \mu(\cdot)) = \int_{\mathbb{R}^2} |x - (2, -2)|^2 d\mu(x)$. The

terminal distribution is set as $\mu_T(x_T) = 1$ to minimize the terminal condition. With these settings, NF-MKV Net trains the MFGs model associated with the dynamic system.

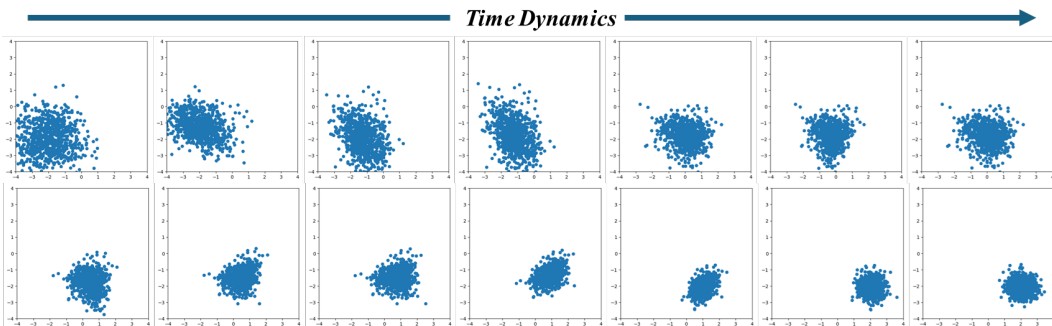

Figure 3: 2-dimensional crowd motion dynamics flow

**d = 50 Crowd Motion.** Similar to the 2-dimensional case, high-dimensional methods adopt the same settings as in Eq.(3) and Eq.(18). In contrast, high-dimensional methods handle agent states and controls in $\mathbb{R}^{50}$, along with density distributions in $\mathcal{L}(\mathbb{R}^{50})$. So Our initial density $\mu_0(x)$ is a Gaussian centered at $(-2, -2, 0, \cdots, 0)$ and terminal conditions $g(x, \mu(\cdot)) = \int_{\mathbb{R}^{50}} |x - (2, 2, 0, \cdots, 0)|^2 d\mu(x)$. Also, the optimal terminal density distribution can be written as $\mu_T(x_T) = 1$. With these settings, NF-MKV Net trains the MFGs model associated with the dynamic system. Results are visualized in the first two dimensions by summing projections from higher dimensions onto these two dimensions.

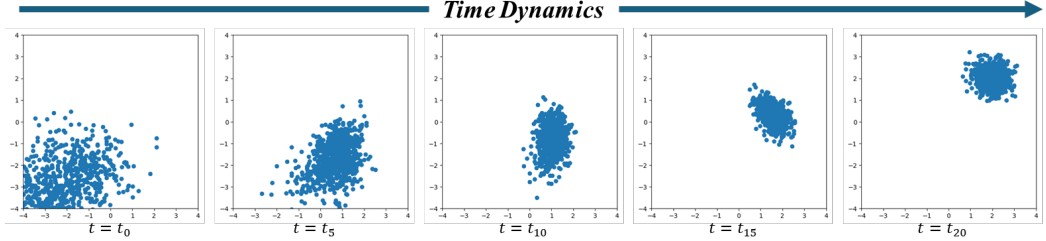

Figure 4: 50-dimensional crowd motion dynamics flow

The trajectories of 1000 points are shown in Fig. (3) for the 2-dimensional case and Fig. (4) for the 50-dimensional case. NF-MKV Net effectively transforms the initial Gaussian density into the terminal condition along a nearly straight trajectory, while ensuring crowd deformation and inter-group collision avoidance. This behavior remains consistent as the dimensionality increases.

### 4.1.3 EXAMPLE 3: MFGS CROWD MOTION WITH OBSTACLE

This experiment considers an MFGs problem with complex process interaction costs. Following the general setting in Eq.(3) and Eq.(18), we change the process interaction costs $f$ as

$$f(x, \mu) = \int_{\mathbb{R}^n} \mathbf{e}^{-|x-\hat{x}|^2} d\mu(\hat{x}) + \int_{\mathbb{R}^n} \left( \mathbf{e}^{-|x-x_o|^2} + \mathbf{e}^{-||x-x_o|^2 - s_{\text{safe}}|^2} \right) d\mu(x). \tag{19}$$

We set $\sigma = \sqrt{2}$ with 20 time steps in the dynamics process and set initial distribution as $\mu_0(x) = \mathcal{N}((-2, -2); 1^2)$. To reach the goal point, we set $x_T = (2, 2)$ which means the terminal condition is $g(x, \mu(\cdot)) = \int_{\mathbb{R}^2} |x - (2, 2)|^2 d\mu(x)$. The terminal distribution should be $\mu_T(x_T) = 1$ to minimize the terminal condition. During the dynamics, the system optimizes the process loss $f$ defined in Eq. (19). The problem involves transforming the initial Gaussian density to a new location while minimizing terminal conditions, avoiding congestion, and bypassing an obstacle at $x_o = (2, -2)$ with a safety radius $s_{\text{safe}} = 1.5$. With these settings, NF-MKV Net successfully trains the MFGs model associated with the dynamics system.

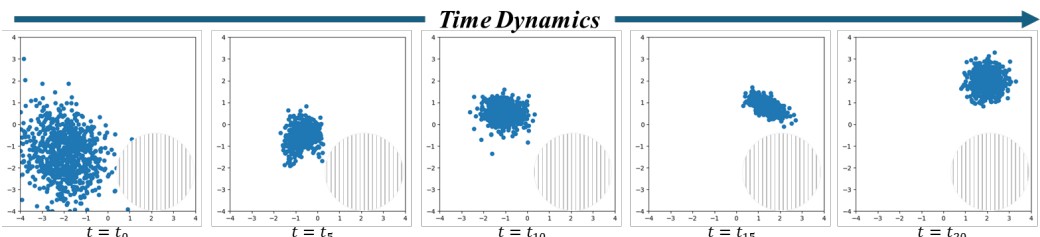

Figure 5: 2-dimensional crowd motion dynamics flow with an obstacle.

The results, shown in Fig. (5), demonstrate that NF-MKV Net successfully transforms the initial Gaussian density into the desired terminal condition along an optimized trajectory, ensuring crowd deformation, inter-group collision avoidance, and obstacle avoidance.

## 4.2 COMPARISON WITH OTHER METHODS

To verify volumetric invariance and time continuity, we compare NF-MKV Net with existing MFGs solving methods, including the distribution-based **RL-PIDL** method proposed by Chen et al. (2023) and the high-dimensional neural network-based **APAC-Net** by Lin et al. (2021).

**Distribution volumetric-invariance.** We implement an approximated integral over the dynamics region, widely used in density estimation such as O'Brien et al. (2016). By generating a grid over a specified area, the numerical integration of a specified probability distribution over that area is computed, and the return value should be close to 1. This method verifies the validity of the distribution. Since the approximated integral is a grid-based method, it can only be used in low-dimensional problems. So, when we approximate integral in $d = 50$ crowd motion, we use the same process as when showing the high-dimensional trajectories from the experiment above, that is, a projection-like method that accumulates the density distribution function on the other components over the first two components and estimates the density in a 2-dimensional region.

**Agents states time-continuity.** The Wasserstein distance is generally chosen as the metric for the difference between two density distributions. Laurière et al. (2022) has used the method to assess the difference between distributions. Even without an explicit probability density function, the Wasserstein distance ($W$-dis) can be computed by optimization methods as long as it can be sampled from both distributions.

We set the comparison experiments in traffic flow ($d = 1$) and crowd motion ($d = 2$ and $50$). The comparison results are shown in Tab.(1) and Fig.(6).

Table 1: Comparison with other methods

|  | **COMPARISON** | **NF-MKV** | **RL-PIDL** | **APAC-NET** |
|---|---|---|---|---|
| Exp 1: traffic flow ($d = 1$) | log of $\mu$ integral difference from 1 | $-\mathbf{2.67}$ | $-0.21$ | / (sample-based) |
|  | $W$-dis of $\mu_t$ between time steps | $\mathbf{0.044}$ | $0.052$ | $0.047$ |
| Exp 2: crowd motion ($d = 2$) | log of $\mu$ integral difference from 1 | $-\mathbf{2.32}$ | $-0.13$ | / (sample-based) |
|  | $W$-dis of $\mu_t$ between time steps | $\mathbf{0.096}$ | $0.108$ | $0.101$ |
| Exp 2: crowd motion ($d = 50$) | log of $\mu$ integral difference from 1 | $-\mathbf{1.06}$ | $0.26$ | / (sample-based) |
|  | $W$-dis of $\mu_t$ between time steps | $\mathbf{2.27}$ | $3.49$ | $2.85$ |

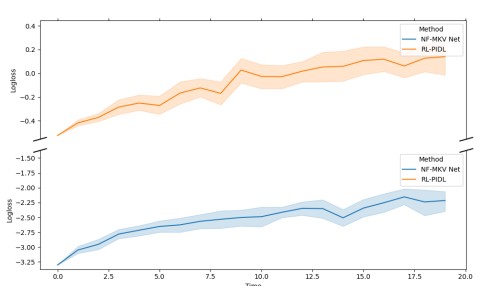 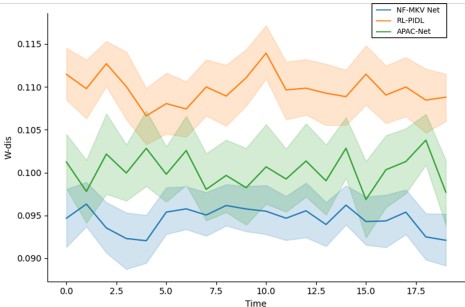

Figure 6: Comparison results in time steps. Left: $\log$ of integration of distribution difference from 1 in traffic flow ($d = 1$). Right: Wasserstein distance of distribution between time steps in crowd motion ($d = 2$).

The **integral error** of the NF-MKV Net method over the dynamic region is below $10^{-2}$ and close to the standard value of 1, significantly outperforming the other method, which has an error exceeding 0.1 in low-dimensional problems. In high-dimensional scenarios, the error of the other method is over ten times larger than that of NF-MKV Net, highlighting the distribution volumetric invariance of NF-MKV Net. The NF-MKV Net has the smallest average **Wasserstein distance** between adjacent time steps among the three methods. This demonstrates smoother evolution and superior agent state time-continuity.

In summary, NF-MKV Net excels in distribution volumetric invariance and agent state time-continuity, making it suitable for solving MFGs problems involving density distributions.

## 5 CONCLUSION

This paper investigates MFGs equilibrium solutions using a stochastic process framework, addressing equivalent probability distribution flow fixed-point problems instead of directly solving the coupled MFGs equations. We propose NF-MKV Net, which integrates process-regularized NF with state-policy-connected time-series neural networks. Process-regularized NF frameworks enforce mathematical constraints by regulating the transfer functions to represent flows of probability measures. State-policy-connected time-series neural networks, grounded in MKV FBSDEs, establish relationships among value functions to ensure a time-consistent process. The proposed method is validated in diverse scenarios, demonstrating its effectiveness compared to existing approaches. NF-MKV Net exhibits strong performance in distribution volumetric invariance and agent state time-continuity, making it applicable to MFGs problems involving distributions.

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

# A  ASSUMPTION(MFGS SOLVABILITY HJB)

(**A1**) The volatility $\sigma$ is independent of the control parameter $\alpha$.

(**A2**) There exists a constant $L \geq 0$ such that:

$$|b(t, x, \mu, \alpha)| \leq L(1 + |\alpha|), \qquad |f(t, x, \mu, \alpha)| \leq L\left(1 + |\alpha|^2\right)$$
$$\left|\left(\sigma, \sigma^{-1}\right)(t, x, \mu)\right| \leq L, \qquad |g(x, \mu)| \leq L$$

for all $(t, x, \mu, \alpha) \in [0, T] \times \mathbb{R}^d \times \mathcal{P}_2\left(\mathbb{R}^d\right) \times A$.

(**A3**) For any $t \in [0, T], x \in \mathbb{R}$, and $\mu \in \mathcal{P}_2\left(\mathbb{R}^d\right)$, the functions $b(t, x, \mu, \cdot)$ and $f(t, x, \mu, \cdot)$ are continuously differentiable in $\alpha$.

(**A4**) For any $t \in [0, T]$, $\mu \in \mathcal{P}_2\left(\mathbb{R}^d\right)$ and $\alpha \in A$, the functions $b(t, \cdot, \mu, \alpha)$, $f(t, \cdot, \mu, \alpha)$, $\sigma(t, \cdot, \mu)$ and $g(\cdot, \mu)$ are $L-$Lipschitz continuous in $x$; for any $t \in [0, T], x \in \mathbb{R}$ and $\alpha \in A$, the functions $b(t, x, \cdot, \alpha)$, $f(t, x, \cdot, \alpha)$, $\sigma(t, x, \cdot)$ and $g(x, \cdot)$ are continuous in the measure argument with respect to the 2-Wasserstein distance.

(**A5**) For the same constant $L$ and for all $(t, x, \mu, \alpha) \in [0, T] \times \mathbb{R}^d \times \mathcal{P}_2\left(\mathbb{R}^d\right) \times A$,

$$|\partial_\alpha b(t, x, \mu, \alpha)| \leq L, \quad |\partial_\alpha f(t, x, \mu, \alpha)| \leq L(1 + |\alpha|).$$

(**A6**) Letting
$$H(t, x, \mu, y, \alpha) = b(t, x, \mu, \alpha) \cdot y + f(t, x, \mu, \alpha)$$

for all $(t, x, \mu, \alpha) \in [0, T] \times \mathbb{R}^d \times \mathcal{P}_2\left(\mathbb{R}^d\right) \times A$, there exists a unique minimizer $\hat{\alpha}(t, x, \mu, y) \in \arg\min_\alpha H(t, x, \mu, y, \alpha)$, continuous in $\mu$ and $L-$Lipschitz continuous in $(x, y)$, satisfying:
$$|\hat{\alpha}(t, x, \mu, y)| \leq L(1 + |y|),$$

for all $(t, x, \mu, y) \in [0, T] \times \mathbb{R}^d \times \mathcal{P}_2\left(\mathbb{R}^d\right) \times \mathbb{R}^d$.

# B  SYMBOLS TABLE

To ensure clarity, we have included a correspondence table for all symbols to enhance the reading experience.

Table 2: Table of symbols

| Symbol | Meaning |
|---|---|
| $\Omega$ | State Process |
| $\mathbb{F}$ | Filtration in the process |
| $\mathcal{F}_t$ | Filtration at time t |
| $\mathbb{P}$ | Probability space measure |
| $\mathbf{W}$ | Wiener Process in the process |
| $W_t$ | Wiener Process at time t |
| $\xi$ | Initial Condition |
| $L^2(\cdot)$ | Square Integrable Function Space |
| $\mathbb{R}^d$ | d-dimensional real number space |
| $\mu$ | Probability Density Measures Flows in the process |
| $\mu_t$ | Probability Density at time t |
| $\alpha$ | Control Action Process |
| $A$ | Control Action Set |
| $J^\mu(\alpha)$ | Loss Functions under Behavioral and Density Flow |
| $\mathbb{E}$ | Expectation |
| $f$ | Process Loss Function |
| $g$ | Terminal Loss Function |
| $b$ | State Control |
| $\sigma$ | Random Perturbation |
| $X_0^\alpha$ | Initial State under a Particular Behavioral Flow |
| $\hat{X}_t^\mu$ | Optimal State of the Representative Agent |
| $L(\hat{X}_t^\mu)$ | The marginal density flow corresponding to the optimum |
| $u$ | Value Function |
| $v$ | Random Perturbation |
| $H(x,p)$ | Hamiltonian function corresponding to state $x$ and covariates |
| $X_t$ | State at time $t$ |
| $Y_t$ | Value Function in MKV FBSDE at time $t$ |
| $Z_t$ | Covariate in MKV FBSDE at time $t$ |
| $r_t$ | Intermediate Conversion Functions for each layer in NF |
| $R(x)$ | Conversion Functions in NF |
| $s_t$ | The inverse of the intermediate conversion function for each layer in the NF |
| $S(x)$ | The inverse of the conversion function of NF |
| $\sigma^{-1\dagger}$ | The inverse ergodic conjugate transpose of sigma |
| $u(0,x|\theta_0)$ | Neural network representation of the value function at time $t=0$ |
| $\partial_x u(x,t|\theta_n)$ | Neural network representation of the gradient of the value function |
| $\hat{u}$ | Final output |
| $\mathbf{P}(g,u)$ | Maximum likelihood of function $g$ and $u$ |
| $l_{MKV}$ | Training Loss of the MKVFBSDE |
| $R_{1,2,\cdots,n}(x;\Phi)$ | Neural network parameter representation of NF |
| $l_{HJB}$ | Training Loss of the HJB equation |
| $N,M$ | Number of time segments, number of full-plane samples |
| $l_T$ | Terminal Loss |
| $r_t(x;\Phi_t)$ | Parameter representation of the NF conversion function |
| $x \sim \mu$ | Sample |
| $p$ | Covariate in Hamiltonian function |
| $u_x$ | $u$ takes a partial derivative with respect to $x$ |

## C  THEORETICAL ANALYSIS

The theoretical work of our designed algorithm are primarily reflected in using the **Representation Theorem for Strong Formulation** to guarantee that the neural network solution corresponds to the equilibrium of Mean-Field Games (MFGs).

In fact, MFGs do not always have an equilibrium; its existence depends on the form of the value function in the equations. According to the conditions for the existence of equilibrium in MFGs[1] and the Representation Theorem for Strong Formulation, the objective function must satisfy the Lipschitz continuous, continuously differentiable and convexity condition (see Appendix A). If an objective function that fails to meet these conditions is directly used as the network's loss function, the resulting solution cannot be guaranteed to correspond to the equilibrium of the MFGs, even if the network provides a solution. At the same time, the solution of the two equations also requires iterative solving, as the MFGs achieves a fixed point through their iteration.

We employ two networks to represent the forward and backward equations, respectively. The value function of each network is strongly tied to the objective function of the equation, following an approach similar to that of Han et al.[2], which effectively represents the equations. And by designing the iterative training structure of the two networks, we characterize the iterative solution process of the two equations in the MKV FBSDE solution process.

These measures theoretically guarantee the existence and uniqueness of the equilibrium in the MFGs system, ensuring that the solution produced by our algorithm is indeed the equilibrium.

In order to show the existence of solutions to the MKV FBSDEs and the equivalence with the MFGs under the given Solvability HJB conditions for the MKV FBSDEs used, we give theoretical proofs which can illustrate the validity of our proposed transformational approach to the equilibrium of the MFGs.

The objective is to prove that, for a given initial condition, the FBSDE has a solution with a bounded martingale integrand, and that this solution is unique within the class of solutions with bounded martingale integrands. Meanwhile, we must also construct a decoupling field. Below shows the theoretical analysis.

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

**Theorem: Representation Theorem for Strong Formulation.**

For the same input $\mu$ as above and under assumption MFGs Solvability HJB, the FBSDE with $X_0 = \xi$ as initial condition at time 0 has a unique solution $(X_t^{0,\xi}, Y_t^{0,\xi}, Z_t^{0,\xi})_{0 \leq t \leq T}$ with $(Z_t^{0,\xi})_{0 \leq t \leq T}$ being bounded by a deterministic constant, almost everywhere for $\mathrm{Leb}_1 \otimes \mathbb{P}$ on $[0, T] \times \Omega$.

Moreover, there exists a continuous mapping $u : [0, T] \times \mathbb{R}^d \to \mathbb{R}$ Lipschitz continuous in $x$ uniformly with respect to $t \in [0, T]$ and to the input $\mu$, such that, for any initial condition $\xi \in L^2(\Omega, \mathcal{F}_0, \mathbb{P}; \mathbb{R}^d)$, the unique solution $(X_t^{0,\xi}, Y_t^{0,\xi}, Z_t^{0,\xi})_{0 \leq t \leq T}$ to the FBSDE with $X_0 = \xi$ as initial condition at time 0, satisfies:

$$\mathbb{P}\left[\forall t \in [0, T], \quad Y_t^{0,\xi} = u\big(t, X_t^{0,\xi}\big)\right] = 1.$$

Also, the process $(\sigma(t, X_t^{0,\xi}, \mu_t)^{-1\dagger} Z_t^{0,\xi})_{0 \leqslant t \leqslant T}$ is bounded by the Lipschitz constant of $u$ in $x$. Finally, the process $(X_t^{0,\xi})_{0 \leqslant t \leqslant T}$ is the unique solution of the optimal control problem. In particular, $\mathbb{E}[u(0, \xi)] = J^\mu(\hat{\alpha})$ for

$$\hat{\alpha} = (\hat{\alpha}(t, X_t^{0,\xi}, \mu_t, \sigma(t, X_t^{0,\xi}, \mu_t)^{-1\dagger} Z_t^{0,\xi}))_{0 \leqslant t \leqslant T}.$$

**Proof.**  We split the proof into successive steps.

*First Step.* We first focus on a truncated version of FBSDE, namely:

$$
\begin{cases}
dX_t = b\big(t, X_t, \mu_t, \hat{\alpha}\big(t, X_t, \mu_t, \sigma(t, X_t, \mu_t)^{-1\dagger}Z_t\big)\big)dt + \sigma(t, X_t, \mu_t)dW_t, \\
dY_t = -\psi(Z_t)f\big(t, X_t, \mu_t, \hat{\alpha}\big(t, X_t, \mu_t, \sigma(t, X_t, \mu_t)^{-1\dagger}Z_t\big)\big)dt + Z_t \cdot dW_t,
\end{cases}
\tag{20}
$$

for $t \in [0, T]$, with the terminal condition $Y_t = g(X_T, \mu_T)$, for a cut-off function $\psi : \mathbb{R}^d \to [0, 1]$, equal to 1 on the ball of center 0 and radius $R$, and equal to 0 outside the ball of center 0 and radius $2R$, such that $\sup |\psi'| \leqslant 2/R$. For the time being, $R > 0$ is an arbitrary real number. Its value will be fixed later on.

By Ref.1, we know that, for any initial condition $(t_0, x) \in [0, T] \times \mathbb{R}^d$, Eq.(20) is uniquely solvable. We denote the unique solution by $(X^{R;t_0,x}, Y^{R;t_0,x}, Z^{R;t_0,x}) = (X_t^{R;t_0,x}, Y_t^{R;t_0,x}, Z_t^{R;t_0,x})_{t_0 \leqslant t \leqslant T}$. Thanks to the cut-off function $\psi$, the driver of Eq.(20) is indeed Lipschitz-continuous in the variable $z$. Moreover, the solution can be represented through a continuous decoupling field $u^R$, Lipschitz continuous in the variable $x$, uniformly in time. Also, the martingale integrand $Z^{R;t_0,x}$ is bounded by $L$ times the Lipschitz constant of $u^R$, with $L$ as in assumption MFGs Solvability HJB. Therefore, the proof boils down to showing that we can bound the Lipschitz constant of the decoupling field independently of the cut-off function in Eq.(20).

*Second Step.* In this step, we fix the values of $(t_0, x) \in [0, T] \times \mathbb{R}^d$ and $R > 0$, and we use the notation $(X, Y, Z)$ for $(X^{R;t_0,x}, Y^{R;t_0,x}, Z^{R;t_0,x})$. We then let $(\mathcal{E}_t)_{0 \leqslant t \leqslant T}$ be the Doléans-Dade exponential of the stochastic integral:

$$
\left( -\int_0^t \big[(\sigma^{-1}b)(s, X_s, \mu_s, \hat{\alpha}_s)\big] \cdot dW_s \right)_{0 \leqslant t \leqslant T},
$$

where $\hat{\alpha}_s = \hat{\alpha}(s, X_s, \mu_s, \sigma(s, X_s, \mu_s)^{-1\dagger}\mathbb{Z}_s)$. As earlier, we write $(\sigma^{-1}b)(t, x, \mu, \alpha)$ for $\sigma(t, x, \mu)^{-1}b(t, x, \mu, \alpha)$ despite the fact that $\sigma^{-1}$ and $b$ do not have the same arguments. Since the integrand is bounded, $(\bar{\mathcal{E}}_t)_{0 \leqslant t \leqslant T}$ is a true martingale, and we can define the probability measure $\mathbb{Q} = \mathcal{E}_T \cdot \mathbb{P}$. Under $\mathbb{Q}$, the process:

$$
\left( W_t^{\mathbb{Q}} = W_t + \int_0^t \big(\sigma^{-1}b\big)(s, X_s, \mu_s, \hat{\alpha}_s)ds \right)_{0 \leqslant t \leqslant T}
$$

is a d-dimensional Brownian motion. Following the proof of Proposition 4.51, we learn that under $\mathbb{Q}$, $(X_t, Y_t, Z_t)_{0 \leq t \leq T}$ is a solution of the forward-backward SDE:

$$
\begin{cases}
dX_t = \sigma(t, X_t, \mu_t)dW_t^{\mathbb{Q}}, \\
dY_t = -\psi(Z_t)f\big(t, X_t, \mu_t, \hat{\alpha}\big(t, X_t, \mu_t, \sigma(t, X_t, \mu_t)^{-1\dagger}Z_t\big)\big)dt \\
\qquad -Z_t \cdot (\sigma^{-1}b)\big(t, X_t, \mu_t, \hat{\alpha}\big(t, X_t, \mu_t, \sigma(t, X_t, \mu_t)^{-1\dagger}Z_t\big)\big)dt \\
\qquad +Z_t \cdot dW_t^{\mathbb{Q}},
\end{cases}
\tag{21}
$$

over the interval $[t_0, T]$, with the same terminal condition as before. Since $Z$ is bounded, the forward-backward SDE Eq.(21) may be regarded as an FBSDE with Lipschitz-continuous coefficients. By the FBSDE version of Yamada-Watanabe theorem proven in ref.1, any other solution with a bounded martingale integrand, with the same initial condition but constructed with respect to another Brownian motion, has the same distribution. Therefore, we can focus on the version of Eq.(21) obtained by replacing $\boldsymbol{W}^{\mathbb{Q}}$ by $\boldsymbol{W}$. If, for this version, the backward component $Y$ can be represented in the form $Y_t = V(t, X_t)$, for all $t \in [t_0, T]$, with $V$ being Lipschitz continuous in space, uniformly in time, and with $Z$ bounded, then $V(t_0, x)$ must coincide with $u^R(t_0, x)$. Repeating the argument for any $(t_0, x) \in [0, T] \times \mathbb{R}^d$, we then have $V \equiv u^R$.

*Third Step.* The strategy is now as follows. We consider the same FBSDE as in Eq.(21), but with $\boldsymbol{W}^{\mathbb{Q}}$ replaced by the original $\boldsymbol{W}$:

$$
\begin{cases}
dX_t = \sigma(t, X_t, \mu_t)dW_t, \\
dY_t = -\psi(Z_t)f\big(t, X_t, \mu_t, \hat{\alpha}\big(t, X_t, \mu_t, \sigma(t, X_t, \mu_t)^{-1\dagger}Z_t\big)\big)dt \\
\qquad -Z_t \cdot (\sigma^{-1}b)\big(t, X_t, \mu_t, \hat{\alpha}\big(t, X_t, \mu_t, \sigma(t, X_t, \mu_t)^{-1\dagger}Z_t\big)\big)dt \\
\qquad +Z_t \cdot dW_t, \quad t \in [0, T],
\end{cases}
\tag{22}
$$

with $Y_T = g(X_T, \mu_T)$, This BSDE may be regarded as a quadratic BSDE. In particular, Ref.1 applies and guarantees that it is uniquely solvable. However, since the driver in the backward equation

is not Lipschitz continuous, we shall modify the form of the equation and focus on the following version:

$$
\begin{cases}
dX_t = \sigma(t, X_t, \mu_t)dW_t, \\
dY_t = -\psi(Z_t)f\big(t, X_t, \mu_t, \hat{\alpha}\big(t, X_t, \mu_t, \sigma(t, X_t, \mu_t)^{-1\dagger}Z_t\big)\big)dt \\
\quad - \psi(Z_t)Z_t \cdot (\sigma^{-1}b)\big(t, X_t, \mu_t, \hat{\alpha}\big(t, X_t, \mu_t, \sigma(t, X_t, \mu_t)^{-1\dagger}Z_t\big)\big)dt \\
\quad + Z_t \cdot dW_t, \quad t \in [0, T].
\end{cases}
\tag{23}
$$

Notice that the cut-off function $\psi$ now appears on the third line. Our objective being to prove that Eq.(23) admits a solution for which $Z$ is bounded independently of $R$, when $R$ is large, the presence of the cut-off does not make any difference.

Now, Eq.(23) may be regarded as both a quadratic and a Lipschitz FBSDE. For any initial condition $(t_0, x)$, we may again denote the solution by $(\boldsymbol{X}^{R;t_0,x}, \boldsymbol{Y}^{R;t_0,x}, \boldsymbol{Z}^{R;t_0,x})$. This is the same notation as in the first step although the equation is different. Since the steps are completely separated, there is no risk of confusion. We denote the corresponding decoupling field by $V^R$. By Theorem in Ref.1, it is bounded (the bound possibly depending on $R$ at this stage of the proof) and $\mathbf{Z}^R; t_0, x$ is bounded.

For the sake of simplicity, we assume that $t_0 = 0$ and we drop the indices $R$ and $t_0$ in the notation $(\boldsymbol{X}^{R:t_0,x}, \boldsymbol{Y}^{R:t_0,x}, \boldsymbol{Z}^{R:t_0,x})$. We just denote it by $(\boldsymbol{X}^x, \boldsymbol{Y}^x, \boldsymbol{Z}^x)$. Similarly, we just denote $V^R$ by $V$.

The goal is then to prove that there exists a constant $C$, independent of $R$ and of the cut-off *functions*, such that, for all $x, x' \in \mathbb{R}^d$,

$$
\left| \mathbb{E}[Y_0^{x'} - Y_0^x] \right| \leqslant C|x' - x|,
\tag{24}
$$

from which we will deduce that, for all $x, x' \in mathbbR^d$

$$
|V(0, x') - V(0, x)| \leqslant C|x' - x|,
$$

which is exactly the Lipschitz control we need on the decoupling field.

*Fourth Step.* We now proceed with the proof of Eq.(24). Fixing the values of $x$ and $x_0$ and letting

$$
(\delta X_t, \delta Y_t, \delta Z_t) = \left( X_t^{x'} - X_t^x, Y_t^{x'} - Y_t^x, Z_t^{x'} - Z_t^x \right), \quad t \in [0, T],
$$

we can write:

$$
d\delta X_t = [\delta\sigma_t \delta X_t]\, dW_t, \quad t \in [0, T],
\tag{25}
$$

where $\delta\sigma_t \delta X_t$ is the $d \times d$ matrix with entries:

$$
(\delta\sigma_t \delta X_t)_{i,j} = \sum_{\ell=1}^d (\delta\sigma_t)_{i,j,\ell}\, (\delta X_t)_\ell, \quad i, j \in \{1, \cdots, d\}^2,
$$

where $(\delta X_t)_\ell$ is the $\ell^{\text{th}}$ coordinate of $\delta X_t$ and

$$
(\delta\sigma_t)_{i,j,\ell} = \frac{\sigma_{i,j}\left(t, X_t^{\ell-1;x\leftarrow x'}, \mu_t\right) - \sigma_{i,j}\left(t, X_t^{\ell;x\leftarrow x'}, \mu_t\right)}{(\delta X_t)_\ell} \mathbf{1}_{(\delta X_t)_\ell \neq 0},
$$

with:

$$
X_t^{\ell;x\leftarrow x'} = \left( (X_t^x)_1, \cdots, (X_t^x)_\ell, (X_t^{x'})_{\ell+1}, \cdots, (X_t^{x'})_d \right).
$$

From the Lipschitz property of $\sigma$ in $x$, the process $(\delta\sigma_t)_0 \leqslant t \leqslant T$ is bounded by a constant $C$ only depending upon $L$ in the assumption. Notice that in the notation $\delta\sigma_t \delta X_t, (\delta\sigma_t \delta X_t)_{ij}$ appears as the inner product of $((\delta\sigma_t)_{i,j,\ell})_{1 \leq \ell \leq d}$ and $((\delta X_t)_\ell)_{1 \leq \ell \leq d}$. Because of the presence of the additional indices $(i, j)$, we chose not to use the inner product notation in this definition. This warning being out of the way, we may use the inner product notation when convenient.

Indeed, in a similar fashion, the pair $(\delta Y_t, \delta Z_t)_{0 \leqslant t \leqslant T}$ satisfies a backward equation of the form:

$$
\delta Y_t = \delta g_T \cdot \delta X_T
$$
$$
+ \int_t^T \left( \delta F_s^{(1)} \cdot \delta X_s + \delta F_s^{(2)} \cdot \delta Z_s \right) ds - \int_t^T \delta Z_s \cdot dW_s, \quad t \in [0, T],
\tag{26}
$$

where $\delta g_T$ is an $\mathbb{R}^d$-valued random variable bounded by $C$ and $\delta \boldsymbol{F}^{(1)} =$ $(\delta F_t^{(1)})_{0 \leqslant t \leqslant T}$ and $\delta \boldsymbol{F}^{(2)} = (\delta F_t^{(2)})_{0 \leq t \leq T}$ are progressively measurable $\mathbb{R}^d$-valued processes, which are bounded, the bounds possibly depending upon the function $\psi$. Here,"" denotes the inner product of $\mathbb{R}^d$. Notice that, as a uniform bound on the growth of $\delta \boldsymbol{F}^{(1)}$ and $\delta \boldsymbol{F}^{(2)}$, we have:

$$
\begin{aligned}
|\delta F_t^{(1)}| &\leqslant C\big(1 + |Z_t^x|^2 + |Z_t^{x'}|^2\big) \\
|\delta F_t^{(2)}| &\leqslant C\big(1 + |Z_t^x| + |Z_t^{x'}|\big)
\end{aligned}, \quad t \in [0, T], \tag{27}
$$

the constant $C$ only depending on the constant $L$ appearing in the assumption and where we used the assumption $\sup |\psi'| \leqslant 2/R$. Since $\delta \boldsymbol{F}^{(2)}$ is bounded, we may introduce a probability $\mathbb{Q}$ (again this is not the same $\mathbb{Q}$ as that appearing in the second step, but, since the two steps are completely independent, there is no risk of confusion), equivalent to $\mathbb{P}$, under which $(W_t^{\mathbb{Q}} = W_t - \int_0^t \delta F_s^{(2)} ds)_{0 \leqslant t \leqslant T}$ is a Brownian motion. Then,

$$
|\mathbb{E}(\delta Y_0)| = \big|\mathbb{E}^{\mathbb{Q}}(\delta Y_0)\big| = \left|\mathbb{E}^{\mathbb{Q}}\left[\delta g_T \cdot \delta X_T + \int_0^T \delta F_s^{(1)} \cdot \delta X_s . ds\right]\right|. \tag{28}
$$

In order to handle the above right-hand side, we need to investigate $d\mathbb{Q}/d\mathbb{P}$. This requires to go back to Eq.(27) and to Eq.(23).

*Fifth Step.* The backward equation in Eq.(23) may be regarded as a BSDE satisfying assumption Quadratic BSDE, uniformly in $R$. By Ref.1, the integral $(\int_0^t Z_s^x \cdot dW_s)_{0 \leq t \leq T}$ is of Bounded Mean Oscillation and its BMO norm is independent of $x$ and $R$. Without any loss of generality, we may assume that it is less than $C$.

Coincidentally, the same holds true if we replace $Z_s^x$ by $\delta F_s^{(2)}$ from Eq.(26), as $|\delta F_s^{(2)}| \leqslant C(1 + |Z_s^x| + |Z_s^{x'}|)$. By Ref.1, we deduce that there exists an exponent $r > 1$, only depending on $L$ and $T$, such that (allowing the constant $C$ to increase from line to line):

$$
\mathbb{E}\left[\left(\frac{d\mathbb{Q}}{d\mathbb{P}}\right)^r\right] \leqslant C.
$$

Now Eq.(25) implies that, for any $p \geqslant 1$, there exists a constant $C_p'$, independent of the cutoff functions $\psi$, such that $\mathbb{E}[\sup_{0 \leqslant t \leqslant T} |\delta X_s|^p]^{1/p} \leqslant C_p'|x - x'|$. Therefore, applying Hölder's inequality, Eq.(28) and the bound for the $r$-moment of $d\mathbb{Q}/d\mathbb{P}$, we obtain:

$$
|\mathbb{E}(\delta Y_0)| \leqslant C|x - x'|\left\{1 + \mathbb{E}\left[\left(\int_0^T \left(|Z_s^x|^2 + |Z_s^{x'}|^2\right) ds\right)^{r'}\right]^{1/r'}\right\},
$$

for some $r' > 1$. In order to estimate the right-hand side, we invoke Ref.(1) again. We deduce that:

$$
|\mathbb{E}(\delta Y_0)| \leqslant C'|x - x'|,
$$

for a constant $C'$ that only depends upon $L$ and $T$. This proves the required estimate for the Lipschitz constant of the decoupling field associated with the system (23).

# D   ERROR ANALYSIS

The error due to discretized MKV FBSDEs is negatively correlated with the number of temporal discretizations $N$, i.e., $O(\delta_u) \sim O(\frac{1}{N})$. Therefore, the denser the temporal discretization splits, the smaller the resulting error in discretizations. Meanwhile, The training loss can be expressed as the solution loss of the discretized MFGs, and the error is caused by the parameterized Neural Network. Below shows the detail of error analysis.

## D.1   ERRORS CAUSED BY DISCRETIZATION OF MKV FBSDES

For convenience, abbreviations will be used in the following error analysis, that is, $f(t, \cdot)$ will represent $f(t, X_t, L(X_t)\sigma^{-1\dagger}Z_t)$.

In MKV FBSDEs, $Y_t$ represents the value function $u_t$. To get the value function, we integrate the second term in eq.(4) in Section 2.1, that is

$$u_t - u_0 = \int_0^t -f(s, \cdot)ds + \int_0^t Z_s dW_s,$$

subtracting the case at $t = t_n$ from the case at $t = t_{n+1}$ gives:

$$u_{t+1} - u_t = \int_{t_n}^{t_{n+1}} -f(s, \cdot)ds + \int_{t_n}^{t_{n+1}} Z_s dW_s,$$

and can be discretized by Euler method, using $f(t, \cdot)$ to represent the average value in the process, and we can get

$$u(x, t_{n+1}) - u(x, t_n) \approx -f(t_n, \cdot)\Delta t_n + [\partial_x u(x, t)]^T \sigma \Delta W_n.$$

where
$$\Delta t_n = t_{n+1} - t_n, \quad \Delta W_n = W_{t_{n+1}} - W_{t_n}$$

The item $[\partial_x u(x, t)]^T \sigma \Delta W_n$ has no error because $W_t$ is a random process, so there is no difference in its value between $\Delta W_t$ and the integral form. The item $-f(t, \cdot)\Delta t_n$ causes the error from the real value $\int_{t_n}^{t_{n+1}} -f(s, \cdot)ds$. The error can be calculated as (represent by $\delta_n$):

$$\delta = \left| -f(t_n, \cdot)\Delta t_n - \int_{t_n}^{t_{n+1}} -f(s, \cdot)ds \right| = \left| \int_{t_n}^{t_{n+1}} f(s, \cdot)ds - f(t_n, \cdot)\Delta t_n \right|$$

By using *First mean value theorem for definite integrals*, $\exists t' \in [t_n, t_{n+1}]$, such that

$$\delta = \left| (f(t', \cdot) - f(t_n, \cdot)) \Delta t_n \right| \leqslant |f_{max} - f_{min}| \Delta t_n \leqslant |f'|_{max} (\Delta t_n)^2$$

In the whole process, since t is discretized into N parts, the error of the whole process (represent as $\sigma_u$) can be obtained as

$$\delta_u \leqslant \sum_{n=1}^N \delta_n = \sum_{n=1}^N |f'_n|_{max}(\Delta t_n)^2 \leqslant |f'|_{max}(\Delta t_n)^2 \cdot \frac{T}{t_n} = |f'|_{max} \cdot \Delta t_n \cdot T$$

resulting that $\delta_u \sim O(\Delta t_n) = O(\frac{1}{N})$.

In summary, the error $\delta_u$ caused by discretization is related to the inverse of the number of discretization division states. Meanwhile, $\forall \varepsilon > 0, \forall |f'|_{max} < M, \exists N \in \mathbb{N}^*$ and $N > \frac{MT^2}{\varepsilon}$ such that $\delta_u < \varepsilon$.

## D.2   ERRORS CAUSED BY PARAMETERIZED DENSITY FLOW AND VALUE FUNCTION

In MFGs, the loss function of the entire system can be written as

$$\inf_{\alpha \in \mathbb{A}} J^\mu(\alpha) \quad \text{with} \quad J^\mu(\alpha) = \mathbb{E}\left[ \int_0^T f(t, X_t^\alpha, \mu_t, \alpha_t)dt + g(X_T^\alpha, \mu_T) \right].$$

For fixed control $\alpha$, the form of the optimization loss function becomes

$$\inf_{\mu} J^{\mu} = \mathbb{E}\left[\int_0^T f(t, X_t, \mu_t)dt + g(X_T, \mu_T)\right] \tag{29}$$

It can be transformed into a parameterized problem for solving neural networks with $\boldsymbol{\Phi}$

$$\inf_{\mu(\boldsymbol{\Phi})} J^{\mu} = \mathbb{E}\left[\int_0^T f(t, X_t, \mu_t)dt + g(X_T, \mu_T)\right] \tag{30}$$

Thus, the loss $J$ consists of two parts, one for process loss and one for terminal loss.

*Process loss.* As a result of discretizing the MKV FBSDEs and constructing the neural network in this way, the process loss changes from the form of an integral to the form of a cumulative sum of the total loss, passed through the network, and can thus be expressed as $l_{MKV}$.

*Terminal Loss.* In the iterative solution of the NF network by a network of fixed-value functions, the terminal loss is calculated by substituting the loss into the terminal-value function g after sampling by $\mu_T$. Thus it can be expressed as $l_T$.

Thus the above $l_{MKV} + l_T$ can represent the total loss of the MFGs $J$.

*Regularization term loss.* The HJB-FPK equations involved in the MFGs still need to be satisfied to hold throughout the process solution, so $l_{HJB}$ is added as a regularization term.

In summary, the sum of the three loss terms during training can be expressed as the solution loss of the discretized MFGs, and since the error due to discretization has already been analyzed, this part of the error is only due to the error caused by the parameterized Neural Network.

