# OpenReview forum: "NF-MKV Net: A Constraint-Preserving Neural Network Approach to Solving Mean-Field Games Equilibrium"
_ICLR.cc/2025/Conference — Submitted to ICLR 2025_

### Official Review · Reviewer_XZLb · 2024-10-26

**Soundness:** 2
**Presentation:** 1
**Contribution:** 2
**Rating:** 3
**Confidence:** 4

**Summary:**

This paper presents a deep learning-based approach to solve mean-field game problems. The proposed approach first models the value function with MKV FBSDEs where the gradient of the value function is parameterized with a neural network. Next, a MAF normalizing flow  architecture is employed for the Fokker-Planck equation and a loss function encode constraints constraints on each density transfer function. The aim is to have time continuity and volumetric invariance constraints satisfied.

**Strengths:**

This work introduces some interesting ideas. Instead of directly addressing the MFG system, it uses a formulation based on stochastic FBSDEs. It also employs two neural networks: one to parameterize the value function and another for the normalizing flow. The normalizing flow enables density and likelihood approximation at the final time.

**Weaknesses:**

1. This work claims that hard constraints are guaranteed; however, these constraints are only enforced using a soft penalty. This should be clarified earlier, in the abstract, and when the authors state their contributions. It would also benefit the paper if the authors explain the implications of using soft penalties rather than hard constraints.
2. The paper claims there are proofs of the algorithm's validity, but I could not find them in the manuscript. The authors should either remove this claim or include them in the main draft or the appendix.
3. The MAF flow is used to generate the Fokker-Planck equation. A continuous normalizing flow automatically satisfies the time continuity constraints as hard constraints. I recommend the authors look at "Bridging mean-field games and normalizing flows with trajectory regularization" by Huang et al. Note continuous NFs are also used in the work "A Machine Learning Framework ..." by Ruthotto et al, which is also cited in this paper.
4. Lack of code for reproducibility - note these can be submitted anonymously as well. For example, the authors could submit an anonymous GitHub repository or a file sharing service. Additionally, the authors could include a brief description of how to run the code in the paper.

Minor:

1. Line 274: "In Section 3.2, we have expressed the value function and its gradient". There is no value function or gradient in section 3.2. The authors should reference the correct section.
2. Line 129: "... is solvable". It would strengthen this paper if the authors cite or show this result.
3. Line 153: "However, a challenge arises in that MFG, unlike Optimal Transport (OT), lacks initial and terminal density distributions. ... This makes it difficult to frame the problem as a complete density evolution problem". I do not believe there is any difficulty here. See point 3 above, where density can be preserved with continuous NFs.

Overall, the manuscript felt like it was hastily written as it had many long complex sentences with grammatical errors, which made it somewhat difficult to read and understand.

For example, in Line 18, "This paper explores the neural network solution of MFGs equilibrium from the perspective of stochastic process, while coupling process-regularized Normalizing Flow (NF) frameworks and state-policy-connected time series neural networks to
solve the McKean-Vlasov type Forward-Backward Stochastic Differential Equations (MKV FBSDEs) fixed point problems, which is equivalent of the MFGs equilibrium." Could be made more clear by instead stating: "This paper explores the neural network solution of MFGs equilibrium from the perspective of stochastic process. The goal is to couple a process-regularized Normalizing Flow with a McKean-Vlasov type Forward-Backward Stochastic Differential Equation for the value function."

Other examples of such sentences include:

2. Line 58: "To summarize..."
3. Line 123: "be in force".
4. Line 133: "We consider..."

**Questions:**

1) what does $-1\dagger$ represent in eqns (4) and (5)?
2) Equation (5) is one of the primary ingredients of this work. Where did this equation come from? Please either show this or cite exactly the reference for this.
3) Line 253: "Multiple f_i are coupled to finally get the function f we need." Is f here the same as the objective function in equation (1)?

---

> ### Author Response · Authors · 2024-11-24
> **Response to Weakness 1 - Hard Constraints**
>
> Dear Reviewer XZLb,
>
> Thank you for your comprehensive and constructive feedback on our manuscript. We appreciate your recognition of the strengths of our work, and we have carefully considered each of your comments and suggestions. We are pleased that you found our approach, including the use of stochastic FBSDEs, neural networks for the value function, and normalizing flows, to be innovative and interesting. Your remarks motivate us to further improve and clarify our manuscript.
>
> **W1. Response to Weakness 1 - Hard Constraints**
>
> In our manuscript, constraint-preserving refers to the volumetric invariance ensured by the Normalizing Flow (NF) during density flow computations and the time continuity maintained in solving the decoupled equations. These properties are guaranteed by the NF’s transformation function and the reformulated McKean-Vlasov FBSDEs (MKV FBSDEs) model. Although the loss functions used involve soft penalties, this approach aligns with MFGs loss functions for process and terminal constraints. Specific details are provided below.
>
> **A. Hard Constraints in Manuscript.** Hard constraints arise from the invariance of density during MFGs training, a property inherent to the NF method. The NF transformation relies on the Jacobian determinant of the coordinates, ensuring volumetric invariance during density transformations. The transformation from MFGs to MKV FBSDEs, along with their discretization, preserves the temporal continuity of the MFGs system, ensuring continuity in the solution process. The preservation of volumetric invariance and temporal continuity constitutes a hard constraint.
>
> **B. Soft Penalty for Loss Function.** Based on our understanding, the soft penalty mentioned in the review refers to the loss function employed during the learning process to optimize neural network parameters. The loss function in MFGs incorporates both process loss and terminal loss to constrain the entire system. The HJB-FPK coupled equations and the reformulated MKV FBSDEs are solved using this loss function. Since the MFGs system is equivalent to the HJB-FPK system of partial differential equations, decoupling to apply hard constraints is challenging. During training, due to the sampling-based coupled solution, existing MFGs equations are solved using soft constraint loss for gradient descent rather than hard constraints for decoupled solutions. We provide an analysis of the error introduced by the loss function in the appendix for clarity (see **Appendix D**).
>
> Thank you for your suggestion. We will clarify this statement in the revised manuscript.

---

> ### Author Response · Authors · 2024-11-24
>
> **W2. Response to Weakness 2 - Theoretical Analysis**
>
> Thank you for your feedback. We recognize the importance of conducting a theoretical analysis. We have analyzed the theoretical result and added the contents to the **Appendix C \& D**, including a theoretical guarantees for the transformation of MKV FBSDEs as well as an error analysis of the method proposed for estimating the equilibrium of MFGs.
>
> **A. Theoretical Analysis.** The theoretical work of our designed algorithm are primarily reflected in using the **Representation Theorem for the Strong Formulation** to guarantee that the neural network solution corresponds to the equilibrium of Mean-Field Games (MFGs).
>
>
> In fact, MFGs do not always have an equilibrium; its existence depends on the form of the value function in the equations. According to the conditions for the existence of equilibrium in MFGs$^{[1]}$ and the Representation Theorem for the Strong Formulation, the objective function must satisfy the Lipschitz continuous, continuously differentiable and convexity condition (see **Appendix A**). If an objective function that fails to meet these conditions is directly used as the network’s loss function, the resulting solution cannot be guaranteed to correspond to the equilibrium of the Mean-Field Game, even if the network provides a solution. At the same time, the solution of the two equations also requires iterative solving, as the Mean-Field Game achieves a fixed point through their iteration.
>
> We employ two networks to represent the forward and backward equations, respectively. The value function of each network is strongly tied to the objective function of the equation, following an approach similar to that of Han et al.$^{[2]}$, which effectively represents the equations. And by designing the iterative training structure of the two networks, we characterize the iterative solution process of the two equations in the MKV FBSDE solution process.
>
> These measures theoretically guarantee the existence and uniqueness of the equilibrium in the Mean-Field Game system, ensuring that the solution produced by our algorithm is indeed the equilibrium.
>
> In summary, we provide theoretical proofs demonstrating the existence of solutions to the MKV FBSDEs and their equivalence to MFGs under the given Solvability HJB conditions. These proofs validate the proposed transformational approach to MFGs equilibria.
>
> **B. Error Analysis.** For error analysis, the discretization error of MKV FBSDEs is inversely proportional to the number of temporal discretizations $N$, i.e., $O(\delta_u) \sim O(\frac{1}{N})$. Thus, finer temporal discretization results in smaller discretization errors. Additionally, the training loss corresponds to the solution loss of discretized MFGs, with errors arising from the parameterized neural network.
>
> *Reference*
>
> *[1] Lasry J M, Lions P L. Mean field games[J]. Japanese journal of mathematics, 2007, 2(1): 229-260.*

---

> ### Author Response · Authors · 2024-11-24
> **Response to Weakness 3 - Novelty Statement (1/2)**
>
> **W3. Response to Weakness 3 - Novelty Statement**
>
> Differences exist between our approach to generating Fokker-Planck (FP) equations and the method described in the referenced study. Our study emphasizes the coupling between the generated Fokker-Planck (FP) flow and the control of a representative agent (HJB equations), rather than directly solving the FP equations. Most studies employing MAF flow for FP generation focus on solving FP equations with process and initial-end-state density constraints. In contrast, our work incorporates the generated FP equations into MKV FBSDEs, addressing the coupled fixed-point problem. Thus, our approach differs from directly solving the FP equations, as we elaborate below.
>
>
> **Huang's work** and **Ruthotto's work** generates a regularized trajectory for the Optimal Transport (OT) problem using Normalizing Flow (NF), without explicitly formulating the problem as equations. While our work seeks the equilibrium of Mean-Field Games (MFGs) by solving a system of stochastic differential equations using coupled Neural Networks, where the form of equations is very important for us to provide theoretical guarantees for the existence and uniqueness of MFGs equilibrium. Additionally, while MFGs and OT share similarities, they differ significantly, particularly in their constraints on terminal losses and terminal states, as detailed below.
>
> **W3.1 Detail of comparison with FP Equation**
>
> While NF equations have been applied to generate FP equations, MFGs equations require the coupled solution of HJB and FP equations. Common FP equations, often diffusion equations without constraints, align closely with stochastic motion processes, making generation models inherently suitable. In the coupled solution process, solving the FP equation is merely the first step. The main challenge lies in coupling it with the HJB equation under constraints, a key difficulty in solving MFGs. In this paper, we reformulate the problem as MKV FBSDEs. This allows the generated FP equations to serve as the marginal distribution in MKV FBSDEs, influencing the control process. In turn, the control process impacts density evolution, leading to the fixed-point problem central to solving MFGs equilibrium. In summary, using NF to solve the FP problem is only the initial step. The subsequent fixed-point problem represents the main challenge in this context.
>
> **A. Target Problem.** In our work, MFGs problems require solving coupled Hamilton-Jacobi-Bellman (HJB) and FP equations, Solving the FP equation is only the initial step in addressing MFGs. The primary challenge is coupling the density flow, derived from the FP equation, with the HJB equation to achieve equilibrium. Consequently, using NF to solve the FP equation is only the initial step; the subsequent fixed-point computation represents the primary difficulty addressed in this study.
>
> **B. Solving Method.** In our approach, we reformulate the problem as MKV FBSDEs. This transformation enables the FP equation to serve as the marginal distribution that influences the control process, which in turn affects density evolution. This interdependence creates a fixed-point problem essential for determining MFGs equilibria.

---

> ### Author Response · Authors · 2024-11-24
> **Response to Weakness 3 - Novelty Statement (2/2)**
>
> **W3.2 Detail of comparison with Huang's and Ruthotto's work**
>
> The main differences between our work and that of Huang et al. lie in the target problem, role of neural networks, training workflow, as detailed below.
>
> **A. Target Problem.** Our work addresses the numerical solution of Mean-Field Game (MFGs) equilibrium, whereas Huang’s work centers on generating trajectories for the Optimal Transport (OT) problem.
>
> MFGs problem describe the strategic decision-making of numerous interacting agents to minimize the loss  (including process loss and terminal loss) of each agent, where each agent’s behavior is influenced by the collective effect of all agents.$^{[1]}$ While Optimal Transport problem focus on determining the most efficient method to transform one density distribution into another while minimizing a cost function.$^{[2]}$
>
> We can easily discover that the optimal transport problem requires a clear understanding of the terminal density distribution at the outset. Meanwhile, MFGs induce density evolution through terminal loss, making it challenging to determine the final density at the outset.
>
> Of course, we also noticed that Huang’s work proposed an MFGs solution based on their optimal transport framework, incorporating process loss into the network and using the KL divergence with the terminal density as the terminal loss. It is clear that their solution cannot effectively handle complex terminal losses (e.g., those involving terminal control), whereas our method naturally does not have such problems.
>
>
> **B. Role of Neural Networks.** In Huang's \& Ruthotto's work, Neural Network serves as a generator for the optimal transport problem's regularized trajectories, and their work can be summarized as directly designing a network from the problem.
>
> In our work, neural networks are numerical solvers for the two equations of the mean field game, and our work can be summarized as transforming the problem into equations and designing solver networks based on those equations.
>
> The process of our work is somewhat complex, but it has a closer connection to the theory of mean field games.
>
>
> **C. Training Workflow.** All of our work employ normalizing flows; however, we utilize an additional network coupled with normalizing flows, leading to distinct workflows.
>
> Huang's \& Ruthotto's work requires a clear terminal distribution at the beginning, allowing direct application of normalizing flows to map the initial distribution to the terminal one.
>
> In contrast, our work starts without a predefined terminal distribution and relies solely on terminal costs. To make our method usable with normalizing flows, we first need to randomly generate a terminal distribution (which obviously will not be the real terminal distribution). After characterizing the entire random evolution process with normalizing flows, we then use another network to solve for the responding value function and guide the normalizing flow to learn in the direction of reducing the value function. Once both networks converge, the terminal loss derived from the normalizing flow represents the true terminal loss.
>
>
> Therefore, there are differences between Huang's \& Ruthotto's  work and ours. These differences highlight the novelty of our work, which establishes a methodology to address the coupled framework of MFGs equations.
>
> *Reference*
>
> *[1] Lasry J M, Lions P L. Mean field games[J]. Japanese journal of mathematics, 2007, 2(1): 229-260.*
>
> *[2] Peyré G, Cuturi M. Computational optimal transport: With applications to data science[J]. Foundations and Trends® in Machine Learning, 2019, 11(5-6): 355-607.*

---

> ### Author Response · Authors · 2024-11-24
> **Response to Weakness 4&Minor**
>
> **W4. Response to Weakness 4 - Reproducibility**
>
> Thank you for your valuable feedback. We understand the importance of providing code for reproducibility. If our paper is accepted, we will make the code available in the camera-ready version. Additionally, we will include a brief description of how to run the code to ensure clarity and reproducibility.
>
> **W5. Response to Weakness Minor - Presentation Problems**
>
> Thanks for your suggestion. We have invited a friend of us who is a native English speaker to help polish our article. And we hope the revised manuscript could be acceptable for you.
>
> Thank you again for the interesting feedback on our work! If you are satisfied with our answers and the modifications made to the paper, we kindly ask you to consider raising your score.

---

> ### Author Response · Authors · 2024-11-24
> **Response to Question 1-3**
>
> We thank you for your insightful questions and constructive feedback. Below, we address each question in detail.
>
> **Q1. What does $-1\dagger$ represent in eqns (4) and (5)?**
>
> $-1\dagger$ denotes the adjoint operator corresponding to the inverse of the differential operator. This notation emphasizes that the adjoint operation is performed after inversion. $\sigma^{-1\dagger}$ refers to the inverse ergodic conjugate transpose of $\sigma$. In the probabilistic approach, $z\sigma^{-1}$ replaces the dual variable $y$ in the coefficients $b$ and $f$, and consequently in the Hamiltonian $H$ and the minimizer $\hat{\alpha}$. In MKV FBSDEs, the martingale term is represented as $Z_t dW_t$ to reflect that the backward equation is one-dimensional.
> Alternatively, $Z_t$ is viewed as a $d$-dimensional vector, but as a $1\times d$ random matrix when expressed as $Z_t dW_t$. This observation supports the dependence of the coefficients on $\sigma^{-1\dagger}Z_t$ rather than $Z_t\sigma^{-1}$.
>
> **Q2. Equation (5) is one of the primary ingredients of this work. Where did this equation come from? Please either show this or cite exactly the reference for this.**
>
> \textbf{A2:} Equation (5) results from applying the adjoint method alongside the variational principle. Specifically, it is derived as the Euler-Lagrange equation for the functional in equation (3). The details are outlined below.
>
> For simplicity, abbreviations will be adopted in the subsequent error analysis, where $f(t,\cdot)$ denotes $f(t,X_t,L(X_t)\sigma^{-1\dagger}Z_t)$.
>
> In MKV FBSDEs, $Y_t$ represents the value function $u_t$. To get the value function, we integrate the second term in eq.(4) in Section 2.1, that is
> $$du_t = -f(t,\cdot)dt+Z_t\cdot dW_t,$$
>
> integrating both sides with respect to time t, we can get
>
> $$u_t = \int_0^t -f(s,\cdot) ds + \int_0^t Z_sdW_s + u_0$$
>
> in which $u_0$ denotes the initial value of the function $u$. So we have the eq.(5), that is
>
> $$u_t-u_0=\int_0^t -f(s,\cdot) ds + \int_0^t Z_sdW_s,$$
> subtracting the case at $t=t_n$ from the case at $t=t_{n+1}$ gives
>
> $$u_{t+1}-u_t=\int_{t_n}^{t_{n+1}} -f(s,\cdot)ds + \int_{t_n}^{t_{n+1}} Z_sdW_s,$$
> and can be discretized by Euler method, using $f(t,\cdot)$ to represent the average value in the process, and we can get
>
> $$u(x,{t_{n+1}})-u({x,{t_n}}) \approx -f(t_n,\cdot)\Delta t_n+[\partial_x u(x,t)]^T\sigma \Delta W_n.$$
>
> where
>
> $$\Delta t_n=t_{n+1}-t_n,\quad \Delta W_n=W_{t_{n+1}}-W_{t_n}$$
>
> **Q3. Line 253: "Multiple $f_i$ are coupled to finally get the function $f$ we need." Is $f$ here the same as the objective function in equation (1)?**
>
> We apologize for the confusion caused; the two terms are not equivalent. The two are not the same. The function in equation (1) refers to the process cost function in the HJB equation, which is also included in $J$. In contrast, the $f$ in line 253 denotes the function from the NF transformation, derived by coupling multiple $f_i$. To clarify, we have corrected the symbol misuse in the manuscript and included a correspondence table in **Appendix B** for reference.
>
> Thank you again for the interesting feedback on our work! If you are satisfied with our answers and the modifications made to the paper, we kindly ask you to consider raising your score.

---

> > ### Comment · Reviewer_XZLb · 2024-11-26
> >
> > I thank the authors for their effort in addressing the comments and revising the paper.
> >
> > Regarding the note on "handling complex target distributions (e.g., those involving terminal control)", it is unclear how this aligns with the terminal condition in Equation (3), as this would require g(x,T) to also depend on u. If this capability is indeed possible, it would be compelling to see this demonstrated experimentally and theoretically.
> >
> > As the proof was added during the review process (and given the limited time available), I was unable to fully verify its correctness. Nevertheless, its inclusion is important for the paper’s development.
> >
> > Based on these discussions, I would update my score to a 4 if it were feasible, though it does not appear to be the case.

---

> > > ### Author Response · Authors · 2024-11-28
> > >
> > > Dear Reviewer XZLb,
> > >
> > > Thank you for your positive feedback! We appreciate your improved the score of our work. Regarding your uncertainty about terminal constraints in Mean-Field Games (MFGs), we would like to provide several examples.
> > >
> > > In Optimal Transport (OT), the terminal density distribution is often predefined or explicitly specified at the outset. In contrast, MFGs frequently involve scenarios where the terminal density depends on control variables or is influenced by complex loss functions. This dependency introduces significant challenges, as the terminal density distribution cannot be directly expressed or predetermined, making the problems not easy to carve through OT. This issue can be understood through two key factors: the uncertainty in the terminal density distribution and the multiple variables jointed terminal condition. We will discuss it in detail in the two aspects and three typical MFGs scenarios.
> > >
> > > **A1. Uncertainty.** In some MFGs setups, the density distribution is guided by terminal functions, which is unknown initially at the setup and needs to be figured out coupled with individuals. When the terminal loss is linked to the unknown terminal density, it must be iteratively solved to achieve Nash equilibrium. **A2. Multiple Variables.** The terminal value function depends not only on the terminal density but, in some complicated problem settings, also on the control variables, individual value function, or even the states of other groups.
> > >
> > > **B1. Traffic Flow.** Section 5.2 of [1] presents a traffic flow example within the MFG framework. In [MFG-NonSeparable], the terminal cost is expressed as $g(x,T)=(\rho(x,T)+u(x,T)-1)^2.$ In this example, the terminal value function depends not only on density distribution but also on control variables.
> > >
> > > Sometimes, the terminal cost can also be set as a fixed constant, where the focus is instead on studying the dynamic evolution of traffic flow through the process loss. For example, in the [MFG-LWR] framework [2], the terminal cost $g$ equals zero.
> > >
> > > **B2. Opinion Evolution.** In Section III-A in Reference [3] as an example, the authors examine the multi-group opinion evolution through linear-quadratic MFGs (LQ-MFGs), with the terminal value function given as
> > >
> > > $$G_{CNMF}=x_{im}^T(T)\overline{Q_m}x_{im}(T)-2\left(\overline{\Phi_{NN}}+\overline{\Phi_{IC}}\right)x_{im}(T),$$
> > >
> > > where
> > >
> > > $$\overline{\Phi_{NN}}=\overline{s_m^T}\overline{Q_{Im}}+\sum_{n=1}^M\left(\overline{\Gamma_m^n}\overline{x^n}(T)+\overline{\eta_m^n}\right)^T\overline{Q_m^n},$$
> > >
> > > and,
> > >
> > > $$\overline{\Phi_{IC}}=\left[\left(I-\overline{\Gamma_m^m}\right)\overline{x^m}(T)-\overline{\eta_m^m}\right]^T\overline{Q_m^m}\overline{\Gamma_m^m}.$$
> > >
> > > In LQ-MFGs, the density distribution is commonly represented by $\bar{x}$, the mean state of all agents. This is reflected in the formulation through $\overline{\Phi_{NN}}$ and $\overline{\Phi_{IC}}$.
> > >
> > > The value function in this problem captures the relationship between the density distribution of Intra-Population and Inter-Population groups and individual states, rather than solely describing the final group density. Additionally, the terminal group density distribution is generally **unknown** and evolves as the system progresses, converging to the Nash equilibrium.
> > >
> > > **B3. Electrical Network Strategic Pricing.** In electrical network pricing, terminal constraints in MFGs are influenced by the uncertain terminal density distribution. Reference [4] models strategic bidding by energy consumers within the MFG framework, with the terminal value function given as
> > >
> > > $$\sigma\left((E_{i}^{d}-\overline{E}^{d})^{2}+(E_{i}^r-\overline{E}^{r})^{2}\right),$$
> > >
> > > where the terminal constraint reflects the relationship between individual states and the group's density distribution. The authors aim to align individual states with the group’s mean density distribution at the terminal time. Similar to the opinion evolution problem, the mean group density at the terminal time is initially **unknown** and evolves dynamically with the system.
> > >
> > > Thank you again for the interesting feedback on our work! If you are satisfied with our answers, we kindly ask you to consider raising your score.
> > >
> > > *Reference*
> > >
> > > *[1] Huang, Kuang, et al. A game-theoretic framework for autonomous vehicles velocity control: Bridging microscopic diﬀerential games and macroscopic mean field games[J].  Discrete \& Continuous Dynamical Systems-B, 2017.*
> > >
> > > *[2] Assouli M, Missaoui B. Deep learning for Mean Field Games with non-separable Hamiltonians[J]. Chaos, Solitons \& Fractals, 2023, 174: 113802.*
> > >
> > > *[3] Ren L, Jin Y, Niu Z, et al. Hierarchical Cooperation in LQ Multi-Population Mean Field Game With Its Application to Opinion Evolution[J]. IEEE Transactions on Network Science and Engineering, 2024.*
> > >
> > > *[4] Silani A, Tindemans S H. Mean Field Game for Strategic Bidding of Energy Consumers in Congested Distribution Networks[C]//2023 62nd IEEE Conference on Decision and Control (CDC). IEEE, 2023: 3606-3611*

---

### Official Review · Reviewer_iNKU · 2024-10-30

**Soundness:** 2
**Presentation:** 2
**Contribution:** 3
**Rating:** 6
**Confidence:** 2

**Summary:**

Neural network-based methods for solving Mean-Field Games (MFGs) equilibrium are effective in high-dimensional settings but struggle with ensuring the density distribution is one throughout evolutions. This paper addresses this issue by exploring neural network solutions for MFGs equilibrium using a stochastic process approach. It combines process-regularized normalizing flow frameworks with state-policy-connected time series neural networks to solve McKean-Vlasov type Forward-Backward Stochastic Differential Equations, which are equivalent to MFGs equilibrium. The approach converts MFGs equilibrium into MKV FBSDEs, incorporating density distribution into the equations' coefficients within a probabilistic framework, and using neural networks to approximate value functions and gradients. The normalizing flow architectures enforce loss constraints on each density transfer function to ensure the density distribution requirements. Empirical results show better performances than some existing methods.

**Strengths:**

1. The method is novel. Whereas many works try to use a generic neural network to solve physical problems, it is refreshing to see a work designing neural architectures catered towards the problem at hand.
2. The mathematical derivation is sound, and the problem of solving MFGs is interesting.
3. While limited, the numerical experiments are nicely conducted.

**Weaknesses:**

1. Evaluating the loss l_HJB is very expensive, having to compute the Laplacian of the value function u. Is there a way to avoid this?
2. While it is true that normalizing flows preserve the volume of distribution along evolution time, it does not usually achieve SOTA performances like other generative models. Is it possible to use other generative models for this purpose?
3. While the idea is nice and the numerical experiments are thoughtful, they do appear limited, any chance to conduct a more thorough empirical study?

**Questions:**

1. Why is it important to preserve the volume of distributions?
2. Followingly, all flow methods preserve the volume. Generative methods through flow matching, for instance, preserve the volume while achieving SOTA results, why not consider them?

---

> ### Author Response · Authors · 2024-11-24
> **Response to Weakness 1-3**
>
> Dear Reviewer iNKU,
>
> Thank you for your detailed and constructive feedback on our manuscript. We are encouraged by your recognition of the novelty and rigor of our work, as well as the thoughtful suggestions to further strengthen it. Below, we address your comments point by point. Your encouragement motivates us to continue improving this line of research.
>
> **W1. Response to Weakness 1** - Computation of Laplacian
>
> Complex computations have been avoided in our algorithm. The computation of lHJB does require the computation of the Laplacian of the value function u, that is
> $$
> \Delta=\nabla^{2}=\nabla\cdot\nabla=\partial_{x^{1}}^{2}+\ldots+\partial_{x^{n}}^{2}+\partial_{t}^{2}.
> $$
> However, since the value function u is a function of time t as well as the state x, for $\partial_{t}^{2}$ we denote it by the discretization $u_{t_{n+1}}-u_{t_n}$, and for $\partial_{x}^{2}$ we have already represented it using a neural network by $(\partial_xu(x,t)|\theta_n)$, and therefore only need to compute the sum of squares of the two to obtain it.
>
> **W2. Response to Weakness 2** - SOTA of Generative Model
>
> NF preserves volume invariance during loss calculation, which is advantageous despite not achieving SOTA performance. By using terminal density matching as a loss function, the loss can gradually decrease to meet the required range. Incorporating other generative models, such as Continuous NF, could improve flexibility and empirical performance, which we plan to explore in future work.
>
> **W3. Response to Weakness 3** - Empirical Study
>
> The current numerical experiments validate the feasibility and effectiveness of our method. However, we agree that more extensive empirical studies could provide deeper insights. In the revision, we will expand the experiments to include high-dimensional scenarios and complex obstacles to better demonstrate the generality and robustness of our approach.
>
> Thank you again for the interesting feedback on our work! If you are satisfied with our answers and the modifications made to the paper, we kindly ask you to consider raising your score.

---

> ### Author Response · Authors · 2024-11-24
> **Response to Question 1&2**
>
> **Q1. Response to Question 1** - Importance of preserving the volume of distributions}
>
> Preserving the distribution volume is crucial for solving Mean-Field Games (MFGs). This significance can be demonstrated through the MFGs definition and the requirements of specific application scenarios.
>
> **A. Definition.** The fundamental assumptions of mean-field games impose natural constraints, including a density integral equal to 1, the nonnegativity of the density function ($m$), and a hard constraint on volume conservation, as outlined in eq. (1) and eq. (2) of Section 1.2 in the original work on MFGs$^{[1]}$. Many studies using sampling-based models focus on real processes without addressing the population density evolution.
>
> **B. Application Scenario.** In traffic flow problems, the value function often depends on the probability density of an individual’s current state. Maintaining overall density constancy is essential in such cases. Coupled with the MKV-FBSDE equations, individual density flow evolves within the probability density space, ensuring that the total probability remains equal to 1.
>
> *Reference*
>
> *[1] Jean-Michel Lasry and Pierre-Louis Lions. Mean Field Games. Japanese journal of mathematics, 2(1):229–260, 2007.*
>
> **Q2. Response to Question 2** - Other Models
> Thank you for the suggestion. We agree that other models could be considered. However, we chose discrete Normalizing Flow primarily for its volume invariance property, which is essential when coupled with discretized MKV FBSDEs.
> * In the MKV FBSDEs framework, the system evolves using discrete standard flows, which serve as inputs to the marginal distributions to derive the final result. This approach is particularly effective for discrete processes.
> * The Flow Matching method typically trains Continuous Normalizing Flow (CNF). We have conducted experiments on this approach by integrating a continuous standard flow matching model with a density evolution model for MFGs during training.
>
> Thank you again for the interesting feedback on our work! If you are satisfied with our answers and the modifications made to the paper, we kindly ask you to consider raising your score.

---

> > ### Comment · Reviewer_iNKU · 2024-11-24
> >
> > I thank the authors for the detailed response. I will keep my score as is.

---

### Official Review · Reviewer_L9Lu · 2024-11-04

**Soundness:** 2
**Presentation:** 2
**Contribution:** 2
**Rating:** 3
**Confidence:** 4

**Summary:**

This paper introduces a framework for solving mean-field game equilibria by parameterizing the evolution of agent densities using continuous normalizing flows, offering a potentially more scalable alternative to existing methods for addressing the coupled partial differential equations, whether or not deep learning is employed. The authors constrain the probability flow generated by the normalizing flows by minimizing the residuals of the Hamilton-Jacobi-Bellman equation alongside the value function. They empirically validate their proposed algorithm through numerical experiments, comparing it with current methods for solving mean-field games.

**Strengths:**

* This paper presents a novel framework that address the limitations of most existing methods for solving mean-field games, in particular, the scalability to high dimensions and the expressivesness of modeling time-dependent probability densities.

**Weaknesses:**

* The authors assert in the abstract that the paper provides theoretical justifications for their method; however, I have not seen any reporting of the claimed theoretical results.
* The paper includes confusing and self-contradictory notation and expressions. For instance, the authors use the symbol $\phi$ to refer to both the value functions in Section 2.1 and the neural network parameters in Section 3.2. Overall, the presentation is unclear and difficult to follow. I recommend that the authors clean the notations and better organize the paper.
* The empirical validation is insufficient to demonstrate the practical benefits of the proposed algorithm. I recommend that the authors consider incorporating some of the experiments conducted in [1], for example, their synethic 2D examples in Section 4.1, which are more complicated and meaningful navigation tasks than what is reported in this paper. If time permits, I would also recommend that the authors try LiDar experiments reported in [1].

[1] Liu, G. H., Lipman, Y., Nickel, M., Karrer, B., Theodorou, E. A., & Chen, R. T. (2023). Generalized Schr\" odinger Bridge Matching. arXiv preprint arXiv:2310.02233.

**Questions:**

I wonder if the authors could explore connections between the class of mean-field game problems they are considering and the Schrödinger Bridge problems, both of which model processes with fixed initial and terminal conditions. Reformulating their methods within the context of Schrödinger Bridge problems would significantly strengthen the paper.

---

> ### Author Response · Authors · 2024-11-24
> **Response to Weakness**
>
> Dear Reviewer L9Lu,
>
> We sincerely appreciate your thoughtful and detailed review of our manuscript. Your feedback has been invaluable in helping us improve the quality and clarity of our work. We deeply regret the confusion caused by the misuse of symbols in the manuscript. Please accept our heartfelt apologies for the oversight.
>
> **Response to Weakness 1** - Theoretical Analysis
>
> Thank you for your feedback. We recognize the importance of conducting a theoretical analysis. We have analyzed the theoretical result and added the contents to the **Appendix C \& D**, including a theoretical guarantees for the transfomation of MKV FBSDEs as well as an error analysis of the method proposed for estimating the equilibrium of MFGs.
>
> **A. Theoretical Analysis.** The theoretical work of our designed algorithm are primarily reflected in using the **Representation Theorem for the Strong Formulation** to guarantee that the neural network solution corresponds to the equilibrium of Mean-Field Games (MFGs).
>
>
> In fact, MFGs do not always have an equilibrium; its existence depends on the form of the value function in the equations. According to the conditions for the existence of equilibrium in MFGs$^{[1]}$ and the Representation Theorem for the Strong Formulation, the objective function must satisfy the Lipschitz continuous, continuously differentiable and convexity condition (see **Appendix A**). If an objective function that fails to meet these conditions is directly used as the network’s loss function, the resulting solution cannot be guaranteed to correspond to the equilibrium of the Mean-Field Game, even if the network provides a solution. At the same time, the solution of the two equations also requires iterative solving, as the Mean-Field Game achieves a fixed point through their iteration.
>
> We employ two networks to represent the forward and backward equations, respectively. The value function of each network is strongly tied to the objective function of the equation, following an approach similar to that of Han et al.$^{[2]}$, which effectively represents the equations. And by designing the iterative training structure of the two networks, we characterize the iterative solution process of the two equations in the MKV FBSDE solution process.
>
> These measures theoretically guarantee the existence and uniqueness of the equilibrium in the Mean-Field Game system, ensuring that the solution produced by our algorithm is indeed the equilibrium.
>
> In summary, we provide theoretical proofs demonstrating the existence of solutions to the MKV FBSDEs and their equivalence to MFGs under the given Solvability HJB conditions. These proofs validate the proposed transformational approach to MFGs equilibria.
>
> **B. Error Analysis.** For error analysis, the discretization error of MKV FBSDEs is inversely proportional to the number of temporal discretizations $N$, i.e., $O(\delta_u) \sim O(\frac{1}{N})$. Thus, finer temporal discretization results in smaller discretization errors. Additionally, the training loss corresponds to the solution loss of discretized MFGs, with errors arising from the parameterized neural network.
>
> *Reference*
>
> *[1] Lasry J M, Lions P L. Mean field games[J]. Japanese journal of mathematics, 2007, 2(1): 229-260.*
>
> **Response to Weakness 2** - Revised Symbols and Correspondence Table
>
> Following your advice, we reviewed and standardized the symbols used in the manuscript. To enhance clarity, we included a correspondence table for all symbols in the **Appendix B**, aiming to improve both the review process and the reading experience.
>
> **Response to Weakness 3** - Supplementary Experiments
>
> Thank you for your thoughtful review and constructive feedback. Your insights are invaluable for improving our manuscript.
>
> **Regarding Empirical Validation.** We acknowledge the limitations of the current experimental validation in demonstrating the practical benefits of our algorithm. As suggested, we will include experiments with additional obstacles, inspired by the synthetic 2D examples in Section 4.1 of [1], to better showcase the efficacy of our approach in complex navigation tasks.
>
> **LiDAR Experiments.** Incorporating LiDAR experiments from [1] would strengthen validation, but time constraints may prevent full implementation in this revision. We will prioritize this for future work and mention it as a potential extension in the manuscript.
>
> *Reference*
>
> *[1] Liu, G. H., Lipman, Y., Nickel, M., Karrer, B., Theodorou, E. A., \& Chen, R. T. (2023). Generalized Schrödinger Bridge Matching. arXiv preprint arXiv:2310.02233.*
>
> Thank you again for the interesting feedback on our work! If you are satisfied with our answers and the modifications made to the paper, we kindly ask you to consider raising your score.

---

> ### Author Response · Authors · 2024-11-24
> **Response to Questions - Difference from Schrödinger Bridge problems**
>
> Thank you for your insightful comment regarding the potential connection between the class of Mean-Field Games (MFGs) problems we consider and Schrödinger Bridge (SB) problems. We appreciate the opportunity to clarify the distinction between these two frameworks.
>
> **Response to Questions** - Difference from Schrödinger Bridge problems
>
> MFGs and SB problems are similar but very different problems. The main differences between MFGs and SB peoblem lie in the target problem and relationship between agents and population, as detailed below.
>
>
> **A. Target Problem.** Mean-Field Games problem describe the strategic decision-making of numerous interacting agents to minimize the loss (including process loss and terminal loss) of each agent, where each agent’s behavior is influenced by the collective effect of all agents.$^{[1]}$ While the SB problem seeks to find the most likely evolution of a system of particles governed by a stochastic process, such as Brownian motion, that transitions between two given probability distributions over a fixed time horizon.$^{[2]}$
>
> We can easily discover that the SB problem requires a clear understanding of the terminal density distribution at the outset. Meanwhile, MFGs induce density evolution through terminal loss, making it challenging to determine the final density at the outset.
>
> In SB problems, the initial and terminal conditions are defined by density distributions, representing the system’s optimal path from the initial to the terminal density. In contrast, MFGs problems define the initial condition by a density distribution, while the terminal condition is derived by simultaneously minimizing the terminal value function $g$ and the process loss $f$.
>
> **B. Relationship between Agents and Population.** MFGs problems have been extensively studied, and scholars have established the dynamical system equations governing their evolution, namely the HJB-FPK equations. The HJB equation defines the optimal control of the representative agent influenced by the population, while the FPK equation describes the evolution of population density under this control. In MFGs, the coupling of HJB and FPK equations ensures that both the representative agent and the population reach a Nash equilibrium. In contrast, SB problems focus on finding the population’s optimal solution without considering equilibrium or agent interactions. Consequently, SB problems outcomes can differ significantly from MFGs equilibria.
>
> SB problems primarily involve density evolution governed by constrained density equations. In MFGs, solving population density evolution is only the first step. The main challenge is coupling the density flow, from the FPK equation, with the constrained HJB equation to achieve equilibrium. The loss function of MFGs defines the coupled HJB-FPK equations, adding to the difficulty of finding MFGs equilibrium.
>
> In this work, we reformulate the problem using McKean-Vlasov FBSDEs (MKV FBSDEs). This reformulation enables the FPK equation to serve as the marginal distribution that influences the control process in MKV FBSDEs, while the control process simultaneously governs the density evolution. TThis interdependence creates a fixed-point problem crucial for determining MFGs equilibrium.
>
> *Refernce*
>
> *[1] Lasry J M, Lions P L. Mean field games[J]. Japanese journal of mathematics, 2007, 2(1): 229-260.*
>
> *[2] Vargas F, Thodoroff P, Lamacraft A, et al. Solving schrödinger bridges via maximum likelihood[J]. Entropy, 2021, 23(9): 1134*
>
> Thank you again for the interesting feedback on our work! If you are satisfied with our answers and the modifications made to the paper, we kindly ask you to consider raising your score.

---

### Official Review · Reviewer_41gp · 2024-11-08

**Soundness:** 3
**Presentation:** 3
**Contribution:** 3
**Rating:** 6
**Confidence:** 3

**Summary:**

This paper proposes to couple “process-regularized Normalizing Flow (NF) frameworks and state-policy-connected time series neural networks to solve the McKean-Vlasov type Forward-Backward Stochastic Differential Equations (MKV FBSDEs) fixed point problems, which is equivalent of the Mean-Field Games (MFGs) equilibrium,” Empirical and theoretical validations are provided.

**Strengths:**

-	The paper presents detailed introduction and explanation of the proposed method, including how to convert solving a MFG to solving a McKean-Vlasov type Forward-Backward Stochastic Differential Equations (MKVFBSDEs) fixed point problem, how to use a NF to estimate density required, and how to couple the two processes. The paper is compressive, well-structed, and easy to follow.

-	The paper present experiments for cases of traffic flow control, crowd motion, crowd motion with obstacles, and comparisons with RL-PIDL and APAC-Net.

-	I find the paper makes conceptual contributions of proposing a new family of neural networks that could have advantages.

**Weaknesses:**

-	The novelty is not cleared stated and explained. There have been works linking MFG and NF (Huang et al. 2023), and converting MFG to MKVFBSDEs (Carmona et al. 2018). I would suggest to clearly explain the novelty and significance of this paper that exceeds a simple combination of the existing works.

-	The experiments are not sufficient. I would suggest adding (1) experiments of comparison with Deep Generalized Schrödinger Bridge (Liu et al. 2022), Ruthotto et al. (2020), and Chen (2021); (2) experiments in more scenarios, such as opinion depolarisation, path planning problem, etc.

-	Theoretical analysis is nearly not given. I would suggest giving error analysis of using the proposed method to estimate the equilibrium of a mean-field game.

References:

Huang H, Yu J, Chen J, Lai R. Bridging mean-field games and normalizing flows with trajectory regularization. Journal of Computational Physics. 2023 Aug 15;487:112155.

Carmona R, Delarue F. Probabilistic theory of mean field games with applications I-II. Berlin: Springer Nature; 2018.
Liu GH, Chen T, So O, Theodorou E. Deep generalized schrödinger bridge. Advances in Neural Information Processing Systems. 2022 Dec 6;35:9374-88.

Lars Ruthotto, Stanley J. Osher, Wuchen Li, Levon Nurbekyan, and Samy Wu Fung. 2020. A machine learning framework for solving high-dimensional mean f ield game and mean field control problems. Proceedings of the National Academy of Sciences 117, 17 (2020), 9183–9193.

Chen, Yongxin. "Density control of interacting agent systems." IEEE Transactions on Automatic Control 69.1 (2023): 246-260.

**Questions:**

Please address the Weaknesses.

---

> ### Author Response · Authors · 2024-11-24
> **Response to Weakness 1 - Novelty Statements (1/2)**
>
> Dear Reviewer 41gp,
>
> We sincerely thank the reviewer for their thoughtful feedback and constructive suggestions. These comments have helped us better articulate our work.
>
> **Response to Weakness 1** - Novelty Statements
>
> The main distinctions between our work and the referenced studies are summarized below, with detailed explanations provided in Sections 3.1.1 and 3.1.2.
>
> **Huang's work** generates a regularized trajectory for the Optimal Transport (OT) problem using Normalizing Flow (NF), without explicitly formulating the problem as equations. While our work seeks the equilibrium of Mean-Field Games (MFGs) by solving a system of stochastic differential equations using coupled Neural Networks, where the form of equations is very important for us to provide theoretical guarantees for the existence and uniqueness of MFGs equilibrium. Additionally, while MFGs and OT share similarities, they differ significantly, particularly in their constraints on terminal losses and terminal states, as detailed below.
>
> **Carmona's work** indeed provided the transformation relationship between mean field games and MKV FBSDEs, but did not focus on the numerical solution of mean field games. In fact, only simple solution examples of linear-quadratic MFGs was provided. In contrast, our work focuses on addressing more general MFGs cases using MKV FBSDE-guided neural networks.
>
> **W1.1 Detail of comparison with Huang et al. (2023)**
>
> The main differences between our work and that of Huang et al. lie in the target problem, role of neural networks, training workflow, as detailed below.
>
> **A. Target Problem.** Our work addresses the numerical solution of Mean-Field Game (MFGs) equilibrium, whereas Huang’s work centers on generating trajectories for the Optimal Transport (OT) problem.
>
> Mean-Field Games problem describe the strategic decision-making of numerous interacting agents to minimize the loss  (including process loss and terminal loss) of each agent, where each agent’s behavior is influenced by the collective effect of all agents.$^{[1]}$ While Optimal Transport problem focus on determining the most efficient method to transform one density distribution into another while minimizing a cost function.$^{[2]}$
>
> We can easily discover that the optimal transport problem requires a clear understanding of the terminal density distribution at the outset. Meanwhile, MFGs induce density evolution through terminal loss, making it challenging to determine the final density at the outset.
>
> Of course, we also noticed that Huang’s work proposed an MFGs solution based on their optimal transport framework, incorporating process loss into the network and using the KL divergence with the terminal density as the terminal loss. It is clear that their solution cannot effectively handle complex terminal losses (e.g., those involving terminal control), whereas our method naturally does not have such problems.
>
>
> **B. Role of Neural Networks.** In Huang's work, Neural Network serves as a generator for the optimal transport problem's regularized trajectories, and their work can be summarized as directly designing a network from the problem.
>
> In our work, neural networks are numerical solvers for the two equations of the mean field game, and our work can be summarized as transforming the problem into equations and designing solver networks based on those equations.
>
> The process of our work is somewhat complex, but it has a closer connection to the theory of mean field games.
>
>
> **C. Training Workflow.** Both our work and Huang’s employ normalizing flows; however, we utilize an additional network coupled with normalizing flows, leading to distinct workflows.
>
> Huang's work requires a clear terminal distribution at the beginning, allowing direct application of normalizing flows to map the initial distribution to the terminal one.
>
> In contrast, our work starts without a predefined terminal distribution and relies solely on terminal costs. To make our method usable with normalizing flows, we first need to randomly generate a terminal distribution (which obviously will not be the real terminal distribution). After characterizing the entire random evolution process with normalizing flows, we then use another network to solve for the responding value function and guide the normalizing flow to learn in the direction of reducing the value function. Once both networks converge, the terminal loss derived from the normalizing flow represents the true terminal loss.
>
>
> Therefore, there are differences between Huang's work and ours. These differences highlight the novelty of our work, which establishes a methodology to address the coupled framework of MFGs equations.
>
> *Reference*
>
> *[1] Lasry J M, Lions P L. Mean field games[J]. Japanese journal of mathematics, 2007, 2(1): 229-260.*
>
> *[2] Peyré G, Cuturi M. Computational optimal transport: With applications to data science[J]. Foundations and Trends® in Machine Learning, 2019, 11(5-6): 355-607.*

---

> > ### Author Response · Authors · 2024-11-24
> > **Response to Weakness 1 - Novelty Statements (2/2)**
> >
> > **W1.2 Detail of comparison with Carmona et al. (2018)**
> >
> > Carmona et al. focused on analytically solving the linear quadratic mean-field games (MFGs) problem, where the density distribution is explicitly determined number by the expectation value. We address the high-dimensional nonlinear MFGs problem using neural networks to parametrize the density distribution flow, enabling a numerical solution where explicit representation of the density is not feasible. The network modeling theory in our work is inspired by Carmona et al.$^{[1]}$. The primary differences between our work and Carmona’s lie in the expression of density distribution and the solving algorithm, as described below.
> >
> > **A. Density Distribution Expression.** Carmona’s approach assumes that the mean-field term $\mu$ represents the expectation of all individual states and is expressed as a deterministic value. In our work, the mean-field term $\mu$ represents the probabilistic distribution of agents in a given state. It captures the population density and cannot be reduced to a simple expectation. In our framework, $\mu$ exists in a probability density space and must be treated as a probability distribution function. Furthermore, many forms of $\mu$ are challenging to represent as functions, increasing the complexity.
> >
> > **B. Solving Algorithm.** Carmona et al.’s seminal work on MKV FBSDEs mainly tackles MFGs in low-dimensional settings by using Riccati equations to decouple forward and backward equations for analytical solutions. Our approach addresses MFGs by iteratively solving the coupled density distribution flow and value function using intelligent methods. This approach allows us to address the challenges of representing densities in high-dimensional, nonlinear MFGs.
> >
> > This work goes beyond existing frameworks by introducing a novel methodology for solving nonlinear-quadratic MFGs, where the density flow and state variables co-evolve using neural network-based approach. These innovations enable the numerical analysis of high-dimensional, nonlinear MFGs.
> >
> > *Reference*
> >
> > *[1] Carmona R, Delarue F. Probabilistic theory of mean field games with applications I-II[J]. 2018.*

---

> > > ### Author Response · Authors · 2024-11-24
> > > **Response to Weakness 2 & 3**
> > >
> > > **Response to Weakness 2** - Supplementary Experiments
> > >
> > > Thank you for your valuable suggestions. We propose to conduct the following experiments next week.
> > >
> > > **A. Experiments of comparison.** We will compare the log of $\mu$ integral differences from 1 and the Wasserstein distance ($W$-dis) of $\mu_t$ across time steps in three scenarios: traffic flow, 2D crowd motion, and 50D crowd motion. The comparison will involve Deep Generalized Schrödinger Bridge (Liu et al., 2022), Ruthotto et al. (2020), and Chen (2021).
> > >
> > > **B. Experiments in more scenarios.** Based on your comments and feedback from other reviewers, we propose adding a path planning problem in complex multi-obstacle scenarios and an opinion depolarization problem. However, the density distribution of group states is not relevant to the opinion depolarization problem. Thus, we set the density distribution to $\mu (\mathbb{E}_{x_i\in X} (x_i)) = 1$ and use it as the interaction term.
> > >
> > > **Response to Weakness 3** - Theoretical and Error Analysis
> > >
> > > Thank you for your feedback. We recognize the importance of conducting a theoretical analysis. We have analyzed the theoretical result and added the contents to the **Appendix C \& D**, including a theoretical guarantees for the transfomation of MKV FBSDEs as well as an error analysis of the method proposed for estimating the equilibrium of MFGs.
> > >
> > > **A. Theoretical Analysis.** The theoretical work of our designed algorithm are primarily reflected in using the **Representation Theorem for the Strong Formulation** to guarantee that the neural network solution corresponds to the equilibrium of Mean-Field Games (MFGs).
> > >
> > >
> > > In fact, MFGs do not always have an equilibrium; its existence depends on the form of the value function in the equations. According to the conditions for the existence of equilibrium in MFGs$^{[1]}$ and the Representation Theorem for the Strong Formulation, the objective function must satisfy the Lipschitz continuous, continuously differentiable and convexity condition (see **Appendix A**). If an objective function that fails to meet these conditions is directly used as the network’s loss function, the resulting solution cannot be guaranteed to correspond to the equilibrium of the Mean-Field Game, even if the network provides a solution. At the same time, the solution of the two equations also requires iterative solving, as the Mean-Field Game achieves a fixed point through their iteration.
> > >
> > > We employ two networks to represent the forward and backward equations, respectively. The value function of each network is strongly tied to the objective function of the equation, following an approach similar to that of Han et al.$^{[2]}$, which effectively represents the equations. And by designing the iterative training structure of the two networks, we characterize the iterative solution process of the two equations in the MKV FBSDE solution process.
> > >
> > > These measures theoretically guarantee the existence and uniqueness of the equilibrium in the Mean-Field Game system, ensuring that the solution produced by our algorithm is indeed the equilibrium.
> > >
> > > In summary, we provide theoretical proofs demonstrating the existence of solutions to the MKV FBSDEs and their equivalence to MFGs under the given Solvability HJB conditions. These proofs validate the proposed transformational approach to MFGs equilibria.
> > >
> > > **B. Error Analysis.** For error analysis, the discretization error of MKV FBSDEs is inversely proportional to the number of temporal discretizations $N$, i.e., $O(\delta_u) \sim O(\frac{1}{N})$. Thus, finer temporal discretization results in smaller discretization errors. Additionally, the training loss corresponds to the solution loss of discretized MFGs, with errors arising from the parameterized neural network.
> > >
> > > *Reference*
> > >
> > > *[1] Lasry J M, Lions P L. Mean field games[J]. Japanese journal of mathematics, 2007, 2(1): 229-260.*
> > >
> > > Thank you again for the interesting feedback on our work! If you are satisfied with our answers and the modifications made to the paper, we kindly ask you to consider raising your score.

---

### Meta-Review · Area_Chair_vLjU · 2024-12-23

**Metareview:**

This paper proposes a neural network approach, NF-MKV Net, for solving Mean-Field Games (MFG) equilibria by combining normalizing flows with neural networks to solve McKean-Vlasov forward-backward stochastic differential equations. The authors claim their approach is effective for high-dimensional MFG problems.

The paper's primary strength lies in its novel combination of normalizing flows and neural networks for MFG problems, attempting to address limitations of existing methods for high-dimensional cases. However, this novelty is overshadowed by significant weaknesses. The claims of "constraint-preserving" are not adequately justified, and the empirical evaluation is limited, particularly in comparison to state-of-the-art methods. The lack of code for reproducibility further undermines the paper's credibility. Additionally, the presentation is unclear throughout, with confusing notation and insufficient theoretical justification in the original submission.

The decision to reject is based on these critical shortcomings. The inadequate empirical validation fails to convincingly demonstrate the method's superiority or even its effectiveness. The absence of publicly available code severely hinders reproducibility, a cornerstone of scientific research. While the authors attempted to address some concerns in their rebuttal, including adding theoretical analysis, these additions were not sufficient to overcome the paper's fundamental flaws. The combination of limited empirical validation, lack of reproducibility, and unclear presentation renders the paper unsuitable for publication in its current form.

**Additional Comments On Reviewer Discussion:**

The review discussion centered on several key issues: unclear claims about constraint preservation, lack of theoretical justification, limited empirical evaluation, unclear presentation and notation, and lack of code for reproducibility. The authors attempted to address these points in their rebuttal, but their responses were ultimately insufficient.

While the authors added some theoretical analysis and promised additional experiments and code release, these commitments do not address the current shortcomings of the paper. The core issues of inadequate empirical validation, lack of clarity in presentation, and absence of reproducibility remain unresolved. The late addition of theoretical analysis, while potentially valuable, could not be fully verified within the review timeframe.

These factors, combined with the initial deficiencies in the submission, solidify the decision to reject the paper. The potential of the proposed method is overshadowed by the significant work required to bring the paper to an acceptable standard for publication.

---

### Decision · Program_Chairs · 2025-01-22

Reject